# Impacts of Climate Change and Mitigation Strategies for Some Abiotic and Biotic Constraints Influencing Fruit Growth and Quality

**DOI:** 10.3390/plants13141942

**Published:** 2024-07-15

**Authors:** Eunice Bacelar, Teresa Pinto, Rosário Anjos, Maria Cristina Morais, Ivo Oliveira, Alice Vilela, Fernanda Cosme

**Affiliations:** 1Centre for the Research and Technology of Agro-Environmental and Biological Sciences (CITAB), Institute for Innovation, Capacity Building and Sustainability of Agri-Food Production (Inov4Agro), University of Trás-of-Montes and Alto Douro, Quinta de Prados, P-5000-801 Vila Real, Portugal; tpinto@utad.pt (T.P.); ranjos@utad.pt (R.A.); mariacristina.morais@gmail.com (M.C.M.); ivobio@hotmail.com (I.O.); 2Chemistry Research Centre–Vila Real (CQ-VR), Department of Agronomy, School of Agrarian and Veterinary Sciences (ECAV), University of Trás-os-Montes and Alto Douro, P-5000-801 Vila Real, Portugal; avimoura@utad.pt; 3Chemistry Research Centre–Vila Real (CQ-VR), Department of Biology and Environment, School of Life Sciences and Environment, University of Trás-os-Montes and Alto Douro, P-5000-801 Vila Real, Portugal; fcosme@utad.pt

**Keywords:** canopy management, environmental stress, gene editing techniques, integrated pest management, pollinators

## Abstract

Factors such as extreme temperatures, light radiation, and nutritional condition influence the physiological, biochemical, and molecular processes associated with fruit development and its quality. Besides abiotic stresses, biotic constraints can also affect fruit growth and quality. Moreover, there can be interactions between stressful conditions. However, it is challenging to predict and generalize the risks of climate change scenarios on seasonal patterns of growth, development, yield, and quality of fruit species because their responses are often highly complex and involve changes at multiple levels. Advancements in genetic editing technologies hold great potential for the agricultural sector, particularly in enhancing fruit crop traits. These improvements can be tailored to meet consumer preferences, which is crucial for commercial success. Canopy management and innovative training systems are also key factors that contribute to maximizing yield efficiency and improving fruit quality, which are essential for the competitiveness of orchards. Moreover, the creation of habitats that support pollinators is a critical aspect of sustainable agriculture, as they play a significant role in the production of many crops, including fruits. Incorporating these strategies allows fruit growers to adapt to changing climate conditions, which is increasingly important for the stability of food production. By investing in these areas, fruit growers can stay ahead of challenges and opportunities in the industry, ultimately leading to increased success and profitability. In this review, we aim to provide an updated overview of the current knowledge on this important topic. We also provide recommendations for future research.

## 1. Introduction

Climate change is a pressing issue that is already significantly affecting plants that produce edible fruits. Shifts in temperature or extreme temperatures, changes in light intensity, and nutritional imbalances are all factors contributing to the challenges faced by fruit growers today. The quality of fruit is influenced by various factors, with biotic stress caused by living organisms and abiotic stress caused by non-living factors playing a significant role. Biotic stressors like insects, herbivores, nematodes, fungi, bacteria, and weeds can cause damage to fruit crops, leading to changes in their quality and sensory attributes [1]. In contrast, abiotic stressors like temperature extremes and changes in light intensity can impact its nutritional composition and post-harvest suitability [2]. Both types of stress can lead to decreased market value and overall crop yield, underscoring the importance of proper management strategies to mitigate their impact on fruit quality. Farmers must proactively implement strategies to maintain crop yields, fruit quality, and sustainability. Research and innovation in agriculture are crucial for developing solutions to help fruit growers adapt to the changing climate and protect their crops from environmental pressure. This review aims to provide an updated and comprehensive analysis of the challenges faced by several abiotic and biotic constraints in fruit growth and quality, incorporating recent scientific advancements in the field. Furthermore, the importance of proactive measures safeguarding fruit crops from the effects of climate change and other stresses is revised. The review also addresses how recent scientific advancements may help mitigate these challenges.

## 2. Effects of Extreme Temperatures on Fruit Growth and Quality and Strategies to Mitigate These Impacts

Among environmental conditions, temperature is the most crucial factor influencing plant growth and development. It affects various physiological and biochemical processes, including photosynthesis, respiration, water uptake, transpiration, flowering, and seed formation [3]. More extreme temperature events will undoubtedly impact fruit yield [4]. Understanding the effects of extreme temperatures on fruit crops is crucial for ensuring sustainable fruit production and reducing economic losses.

Since 1950, extreme temperature events, including heat waves and cold snaps, have become more frequent and intense globally [5]. Based on the updates available in February 2023 in the Sixth Assessment Report (AR6) of Working Group One (WGI) of the Intergovernmental Panel on Climate Change (IPCC), the global surface temperature has increased by approximately 1.15 °C from the base period of 1850–1900 to the period between 2013 and 2022 [6]. Extreme temperatures on land tend to increase more than the global mean temperature due mainly to the land-sea warming contrast and regional feedback in some regions [5]. As temperatures continue to rise, the global warming trend will dominate further, and thus, frequent extreme heat regimes will expand into higher latitude regions [7]. The increase in global surface temperature, particularly on land, has significant implications for plant growth and development. Fruit crops, in particular, are susceptible to temperature fluctuations, which can produce a range of negative impacts that affect yield and quality. Increases in the frequency and intensity of extreme meteorological events may also promote the spread of pathogens to new locations or evolve to infect other plant species and/or become more virulent [8]. An indirect effect of increasing air temperatures is prolonging the growing season. The length of the growing season and temperatures are likely to influence the crop’s water requirement, which will affect the crop’s water status, especially in arid and semi-arid areas [6]. The extension of the growing season will mostly positively affect fruit crops in the mid-and high latitudes since a longer growing season will improve the scope for cultivar selection. On this point, we explore the effects of extreme temperatures on fruit crops and discuss strategies to mitigate their negative impacts, ensuring a resilient and profitable fruit industry. 

Extremely high day and evening temperatures occurring more frequently now due to climate change can cause the drop of flowers and produce deformed or undersized fruits [7,9]. The significant impact of temperature on fruit growth suggests potential future production challenges. High temperatures, especially when combined with intense sunlight radiation, can lead to sunburn (discussed in Section 5), uneven ripening, altered texture, and decreased nutritional value, thereby reducing the overall appearance and quality of the fruits and compromising their market value. Additionally, dry heat can cause fruit desiccation, shriveling, and browning. To alleviate heat stress effects on fruit set, fruit size, fruit shape, and fruit ripening, farmers can employ strategies such as shade netting [10], moisture management through irrigation, mulching, the use of new heat-tolerant rootstocks and fruit cultivars, application of plant growth regulators (PGRs; natural or synthetic compounds that can regulate developmental and metabolic processes in higher plants), and the use of bagging and film sprays [11]. Additionally, applying melatonin directly to plants or incorporating it into their nutrient regimen can improve their tolerance to temperature stress and enhance fruit development. Some studies have shown that melatonin can increase antioxidant activity in fruits, leading to improved shelf life and overall quality [12].

Farmers can also improve their fruit yields by adopting precision agriculture and artificial intelligence (AI) technologies to monitor and manage the microclimate conditions in their orchards. Farmers must stay informed about the latest research and best practices for managing heat stress in fruit crops to protect their yields and ensure a successful harvest [13]. Conversely, despite the inevitable and permanent consequences of global warming, fruit crops are particularly vulnerable to extreme cold temperatures and frost damage. Freezing temperatures can induce damage to the cell membrane, reduce the activity of scavenging enzymes, destabilize proteins, and disrupt fruit development by delaying flowering and fruit set [14]. Freezing temperatures can ultimately result in reduced sugar content and taste quality. To protect fruit crops from extreme cold and frost damage, farmers can implement measures such as using frost blankets, modifying microclimate, selecting cold-hardy fruit varieties, adopting timely pruning practices, and using PGRs like brassinosteroids or abscisic acid [15]. However, some of the novel systems and materials that have recently been introduced have not yet been thoroughly tested and are not widely embraced by fruit growers.

Drastic temperature fluctuations, particularly during flowering and fruit sets, can also negatively impact fruit crops, resulting in reduced yield and economic losses. Injuries caused by temperature variations include crown necrosis in winter, sunscald during summer, cracking of the xylem and phloem due to frost, and buds’ death during winter [16]. Temperature variations can also increase susceptibility to fungal diseases and pests [17]. Adapting agricultural practices to monitor and manage temperature fluctuations, practicing proper harvest timing and post-harvest techniques, and implementing integrated pest management can help fruit growers mitigate these effects. 

Effects of extreme temperatures during pollination deserve the maximum attention since pollination is the most critical process for fruit set, yield, and quality. Poorly pollinated flowers never become premium fruits. In addition, extreme temperatures can negatively affect the activity of pollinators like bees [18]. Interactions between bees and flowering fruit crop plants require synchrony in their physiology, phenology, and behavior to be productive and successful. Under extreme heat conditions, these interactions are disrupted. This can reduce pollination rates, affecting fruit set, shape, size, and overall fruit crop yield (see Section 6 for more details). 

Additionally, extreme temperature fluctuations can also affect the health and development of the tree, which can, in turn, impact its ability to produce fruit. Overall, while anemophilous pollinated fruit trees may be less directly affected by temperature than insect-pollinated trees, temperature fluctuations can still affect their reproductive success and fruit set. For example, temperature can affect the timing of pollen release and dispersal, as well as the receptivity of the stigma. If temperatures are too high or too low, this can reduce pollination efficiency in anemophilous trees [19].

In a changing climate era, fruit growers must be aware of their fruit crops’ specific temperature requirements and sensitivities and take appropriate measures to protect them. Some fruit crops are more sensitive to extreme temperatures. For example, citrus plants are sensitive to cold temperatures and can experience frost damage [20], reducing fruit quality and yield. On the other hand, vines are sensitive to heat waves, which can lead to sunburn and reduced sugar content in the berries, affecting the wine quality and flavor. Nevertheless, it is essential to mention that in addition to crop susceptibility, all crops can be subject to frost damage, given the intensity, duration, and phenological stage at which the crop is subjected to frost, including grapevine [7].

## 3. Influence of Soil Nutrient Deficiencies or Imbalances on Fruit Growth and Quality and Innovative Approaches to Improve Soil Fertility

Sustainable orchard soil management practices are essential in mitigating the impact of climate change on fruit productivity and quality [21]. The nutrition of fruit trees is crucial for their growth and productivity, and imbalances in soil nutrients can lead to a decrease in fruit quality [22]. To improve and preserve soil resources in the global food system, policies and initiatives must encourage transitions to more nutritious diets, enhance food production intensification strategies and technological advances, restore and preserve soil fertility, and promote greater resilience in climate change [23].

### 3.1. Classification, Definition, and Function of Soil Nutrients in Plant Development

Even in small quantities, soil nutrients are essential for plant development, maturation, fruit quality, and reproduction. These can be divided into macronutrients and micronutrients, and both play a vital role in metabolic and physiological processes. Macronutrients are necessary for the growth and development of plants in high quantities, and micronutrients are essential in small amounts. Furthermore, micronutrients can cause toxicity problems, impacting the production and quality of crops if absorbed in quantities greater than those necessary for each plant species. Currently, it is considered that there are twenty mineral elements essential or beneficial for the growth and development of plants [24]. According to Aftab and Hakeem [25], some nutrients are considered micronutrients because they are needed in small amounts in the soil for the full development of plants and their vitality. When involved in physiological and biochemical processes, these elements contribute to the harmonious development of fruits. Some examples of macronutrients are nitrogen (N), phosphorus (P), potassium (K), calcium (Ca), magnesium (Mg), and sulfur (S) [24,26]. Some micronutrients include iron (Fe), zinc (Zn), manganese (Mn), copper (Cu), molybdenum (Mo), boron (B), nickel (Ni), cobalt (Co), silicon (Si), sodium (Na), and chlorine (Cl).

According to several authors [27,28], these trace elements are indispensable for activating and supporting enzymes that drive vital metabolic reactions.

#### 3.1.1. Macronutrients

Nitrogen is involved in many fundamental physiological processes for plants, being the nutrient needed in higher quantities than others. It is essential for overall vegetative growth, as it is the main element in the composition of proteins, amino acids, hormones, chlorophyll, vitamins, and enzymes crucial for plant life. However, its effects are not limited to growth and productivity but significantly affect the fruit’s quality parameters and antioxidant activity. On the other hand, excess N can adversely affect flowering and plant production [24,29]. The nature and degree of interactive impacts of management practices and site conditions on N use efficiency are not yet well understood, limiting a comprehensive assessment of these practices on N use efficiency. Implementing optimal agricultural management strategies to increase global N efficiency (48%) is urgent to ensure environmental safety and benefits [30].

According to You et al. [31], in all cases studied, nutrient and crop management practices increased N use efficiency, while soil management had the opposite impact and decreased N use efficiency. Crop type, soil pH, soil clay content, soil organic carbon, temperature, and precipitation also affected N use efficiency.

P intervenes in almost all plant growth and metabolism processes. Thus, it is observed that P is translocated to areas with a high energy requirement, such as the fruiting areas of the plant, for the formation of seeds and fruits [24]. P deficiency leads to the purple coloring of leaves and stems and delayed growth and ripening of fruits. It often also causes the premature fall of flowers and fruits, leading to drops in expected production [24].

P is also one of the nutrients that plants use in large quantities and contributes to osmotic regulation and the activation of enzymes involved in sugar metabolism, resulting in juicier and more flavorful fruits. Schachtman and Shin [32] highlight the participation of P in transporting sugars to fruits, directly impacting the flavor and sweetness of agricultural products. It intervenes in stomatal regulation, maintaining the plant’s water balance and promoting stem rigidity and resistance to cold [24]. Additionally, P plays an essential role in plant response to pathogen attack. Adequate P availability in soil can improve plant resistance, positively reflecting fruit production and quality [33].

Ca plays a crucial role in the development and quality of fruits in several plant species by influencing physiological and biochemical processes. The importance of Ca in forming and maintaining the cellular structure of fruits has been the subject of several studies. For example, White and Broadley [34] highlight the role of Ca in regulating the cell wall, helping to maintain the structural integrity of fruits. The adequate presence of Ca contributes to tissue resistance, preventing disorders such as apical rot in tomato plants [35]. Furthermore, it promotes the transport and retention of other elements, influences water movement in cells, and is fundamental to continuous cell growth and division. 

Ca is not very mobile within the plant (it is not translocable), so its absorption is necessary throughout the vegetative cycle [24]. Ca absorption is impaired by excessive amounts of ammonia, N, P, Mg, Mn, and Al in the soil. The presence of B influences the formation and stability of the cell wall, acting as a facilitator of the optimal assimilation and incorporation of Ca into cell walls. Therefore, B deficiency can hinder Ca transport and functionality, reducing fruit growth and shelf life [24].

In addition to its role as a structural component in the cell wall, Ca acts as a second messenger in signal transduction pathways. It is vital for fruit growth as it provides structural support and regulates cellular processes and signaling pathways essential for development and maturation. Another relevant aspect is the influence of Ca on plants’ response to abiotic stress, such as climate variations and soil salinity, which can impact the absorption and translocation of Ca in plants, directly affecting fruit quality and production [36].

Mg is part of the structure of the chlorophyll molecule. Without optimal levels of Mg in the soil, the leaves will lack this macronutrient. Therefore, chlorophyll synthesis decreases, reducing the photosynthetic rate and thus the ability of plants to harness energy from the sun [37], in addition to performing other functions essential to the growth and development of plant species [38]. In some species, Mg deficiency manifests itself through reddening, as in red grape varieties, and other cases, through yellowing of the tissue between the leaf veins due to their low chlorophyll content [24].

S is mainly associated with organic matter, and its deficiency is rare in soils where organic matter is present at adequate levels [39]. This macronutrient is involved in several biochemical processes of plant species, increasing their productivity [39]. Furthermore, S in plants is also involved in the biosynthesis of glucosinolates, a group of nutritionally significant plant secondary metabolites possessing health-beneficial properties [40] and in seed production [24].

#### 3.1.2. Micronutrients

In Figure 1, some micronutrients and their action on plants can be seen.

Fe is essential in various enzymatic reactions, such as its intervention in chlorophyll forming, some proteins, and respiratory function. Due to its limited mobility in the soil, Fe must be close to the roots for easy absorption by plants [41]. This micronutrient is fundamental in metabolic processes such as N metabolism, photosynthesis, and respiration [42]. Reduced quantities in the soil will disrupt these processes, negatively influencing plant vitality and affecting fruit quality [42,43]. Furthermore, micronutrients, such as Zn and Mg, are essential in small quantities for various functions; they are necessary for photosynthesis and chlorophyll production, besides performing other tasks in the plant, particularly at the enzymatic level, which directly influence fruit quality [24,44]. Cu also plays a fundamental role in chlorophyll production and enzymatic activity. B is essential in maintaining the integrity of cell walls. Moreover, several studies report that the presence of B in adequate quantities is necessary for the flowering and fruiting processes and the metabolism of carbohydrates and proteins [45,46]. As it is not very mobile in the plant, the death of terminal buds manifests B deficiency, the formation of short internodes, and the deformation of the leaves. The fruits become deformed and have no commercial value [24].

Co and Mo are essential for symbiotic nitrogen fixation, and Si, a constituent of cell walls, increases the tolerance of some species to heat and dryness, as well as resistance to insects and infections caused by fungi, hereby enhancing plant yield [24]. Furthermore, Mo is necessary for practical protein synthesis [47]. On the other hand, if the soil has Mo deficiencies, plant growth, and productivity can be inhibited [48,49]. Table 1 provides a comprehensive summary of micronutrient deficiencies and their symptoms.

So, it is essential to ensure a balanced and sufficient presence of these nutrients in the soil to promote healthy fruit growth, increase their resistance to diseases, improve storage capacity, and improve nutritional quality. Farmers who value soil health and plant nutrition apply management strategies to optimize these components, resulting in more abundant harvests and higher-quality fruit [24].

### 3.2. Innovative Approaches to Soil Fertility Enhancement

Intensive agricultural practices have left a lasting mark on soil health, negatively impacting microbial life, mineral and organic composition, structure, and physicochemical properties [61]. In response to the challenge of improving soil fertility, innovative and sustainable approaches have emerged, such as adopting new agricultural practices, fertilization methods, sources of correctives, and strategies for plant nutrition. Furthermore, the strategic application of bacteria and fungi has been explored to optimize nutrient use efficiency by crops. These solutions have the main objective of preserving the vitality of the soil, restoring its compromised fertility, and ensuring that plants receive the nutrients necessary for abundant, high-quality agricultural production. This set of practices and innovations actively contributes to promoting the adoption of sustainable farming methods, aligning with sustainable agriculture principles. The emphasis is on balancing efficient food production and long-term soil health conservation, ensuring that future generations can grow food sustainably and responsibly [61,62,63,64,65].

According to Sahoo, Bhardwaj, and Tuteja [66], biofertilizers are microbial inoculants that consist of specific microorganisms, organic substances, and dead plant tissues derived from plant roots and the rhizosphere. The use of these compounds collectively contributes to an attractive and environmentally sustainable strategy to increase crop yields and, at the same time, mitigate dependence on chemical fertilizers. This happens because of the sustainable enrichment of soil fertility as they exhibit beneficial activities, including N fixation, P solubilization/mobilization, phytohormone production, K solubilization, and protein synthesis. Inoculating beneficial microbes, individually or in consortia, is not only a sustainable solution from an environmental perspective but has proven effective in improving crop yields and plant biomass. Other advantages that can be highlighted are their economic viability, non-toxicity, and resistance to pests and diseases [67,68]. Increased awareness among farmers about the positive aspects of biofertilizers has led to their widespread adoption in agricultural practices [69].

Soil compaction reduces its water retention capacity, increasing the risk of erosion and loss of nutrients. To deal with this problem, a technical approach involving bioengineering is essential. Using plants with deep root systems, such as grasses and legumes, plays a crucial role in stabilizing soil, preventing erosion, and maintaining soil structure. Also, crop rotation and cover cropping are effective practices for improving soil structure and minimizing compaction, diversifying crops, and protecting soil during non-crop periods [70]. In a study carried out by the authors just mentioned, they concluded that the combination of crop rotation and irrigation strategies seemed promising for improving the adaptability of agrosystems in the semi-arid region. The addition of organic matter should also be considered as a strategy within the scope of regional adaptation to climate change, as it had a generally positive effect on soil apparent density and water retention capacity, allowing the redistribution of surpluses from water stations [71]. 

In short, progress is achieved by improving agricultural techniques, especially by explicitly addressing the nutritional requirements of plants and considering physiological and environmental conditions. During this process, it is essential to study and evaluate the nutritional quality of the final product, such as food, to meet human needs. This contributes to maintaining fruits as rich sources of minerals in the diet, promoting the population’s health [72,73].

## 4. Strategies for Integrated Pest Management

Agriculture plays a crucial role in ensuring the right to food security, which implies the availability and stability of food supply, as well as the physical, social, and economic accessibility of food of adequate quality and quantity to meet the nutritional needs of people, essential for their survival and well-being [74]. However, agriculture faces new challenges due to climate change, which requires it to adapt to new production conditions and increase food production while ensuring food quality and safety. Furthermore, agriculture is expected to minimize its negative environmental impacts and produce public goods such as environmental, social, and economic benefits not compensated by the market [74]. This multifunctionality of agriculture includes maintaining biodiversity, occupying and preserving rural areas, and fostering prosperous rural communities. One of the main challenges of achieving global food security in the 21st century is ensuring enough food supply to cater to a growing population while conserving natural resources [74]. These two objectives often conflict with agricultural productivity and conservation being viewed as a trade-off. The idea of “feeding the world” is frequently used to justify the excessive use of pesticides and insurance-based pest management approaches in crop protection [74].

Integrated Pest Management (IPM) is an effective process that can control pests while minimizing environmental and human risks and is adaptable to environments, including urban, agricultural, wild, or natural areas [75]. Nevertheless, this crop protection solution is far from universal—it is often tricky, indeed sometimes impossible to implement, due to several factors such as pest dynamics, host–plant and climate interactions, practicalities of crop production, and socioeconomic conditions in the region of interest, which can hinder the full implementation of this strategy [76]. 

The main focus of IPM is to prevent pests or reduce their damage in the long term by using ecosystem-based strategies such as biological control, cultural practice modification, habitat manipulation, and the use of resistant varieties [75]. Implementing IPM strategies ensures a proactive approach to pest control, which is both sustainable and safe [75]. Pesticides are used only when monitoring indicates that they are necessary. The treatments are carried out in a targeted manner to remove only the specific organism causing the problem [77].

Selecting and applying pest control materials minimize risks to human health, beneficial and non-target organisms, and the environment [77]. 

The method IPM emphasizes preventing pests from causing problems rather than eliminating them after they have appeared. This can be achieved in several ways, such as growing healthy crops that can resist pest attacks, using disease-resistant plants, or sealing cracks to prevent insects or rodents from entering a building [77]. The IPM considers environmental factors that can affect the pest’s ability to thrive instead of solely focusing on eliminating the visible pests. By using this information, unfavorable pest conditions could be established, decreasing the chances of infestation. The monitoring process involves examining a field, landscape, forest, building, or any other site to determine the type and quantity of pests present and the damage they have caused [78]. Implementing non-contact, highly efficient, and affordable methods for detecting and monitoring plant diseases and pests over vast areas could greatly facilitate plant protection. In this respect, different remote sensing methods have been introduced for detecting and monitoring plant diseases and pests in many ways [79].

If control is necessary, detailed information about the pest helps select the most effective management techniques and determine the optimal time to apply them. The most successful and sustainable approach to pest management involves combining methods that complement each other. Biological control, habitat manipulation, and legal control can be used together to manage pest populations. Host resistance can be used alone or in combination with these tactics. Chemical control is usually compatible with host resistance. Therefore, an effective management strategy combines one or more compatible tactics into a single package [80].

### 4.1. Biological Control

Biological control, or biocontrol, involves using natural enemies of pests to prevent damage [81]. The term “bio-rationals” [82] has been used as an operative expression to refer to specific components in plant protection strategies, which are assumed to have advantages regarding risk characteristics. At the same time, they provide acceptable efficacy in reducing pest impact. The products are often biologically derived; however, if synthetic, they are structurally similar and functionally identical to a biologically occurring material [82].

Beneficial organisms such as predators, parasitoids, and pathogens are often employed to control pest species and reduce the negative impact of synthetic pesticides on the environment, non-target organisms, and human health. These natural enemies can also include competitors [81].

Biological control involves using bio-based products such as pheromones, resistant plant varieties, and autocidal techniques that include sterile insects [81,82]. Biological control is the most adopted sustainable pest control strategy in agriculture [83].

There are three commonly used methods for applying biological control in agroecosystems: conservation biological control (CBC), classical biological control (BC), and augmentative biological control (ABC). The CBC involves preserving or favoring the existing natural enemies in the ecosystem. The BC introduces new natural enemies to establish a permanent population within the new agroecosystem. Lastly, ABC relies on massive and periodic releases of biocontrol agents to quickly reduce pest populations [83]. The successful implementation of ABC programs relies on CBC practices, which aim to create more balanced and sustainable agroecosystems [83].

The CBC approach involves preserving and enhancing natural biological control by maintaining a habitat that can support natural enemies [84]. This does not necessarily exclude other pest management methods or using different biological control agents through ABC strategies [85]. Instead, CBC helps to create a more favorable environment for biocontrol agents to thrive, resulting in more effective ABC practices [86].

Incorporating some practices of CBC and ABC into IPM can improve the effectiveness of biological control methods. Using BC methods, such as CBC and ABC, as a part of an IPM program can help reduce the reliance on pesticides [87].

When considering the scope of use and comparable efficacy of biological and chemical plant protection products, it is recommended that biological products be prioritized in IPM practices. However, the farmers’ cost and knowledge level must also be considered when making this decision [76].

Many economic studies examining the advantages of pesticides compare two situations: current usage versus complete elimination. These studies investigate the costs farmers and the agricultural industry would incur if pesticide use were reduced. They have concluded that significant expenses would arise due to lower crop yields, the necessity of importing more food, and more substantial health hazards to consumers via exposure to mycotoxins in food [88].

Farmers should be aware that non-chemical methods are available and can be used as an alternative to pesticides. They need to be convinced that these methods are cost-effective. Chemical companies conduct several training sessions for farmers and advisors to educate them on how to use new pesticides and remind them about the known ones [82].

It is important to educate farmers about biological methods and their effectiveness. Microbial and biological products must be appropriately applied, which can be more challenging than chemical products but is crucial for optimal efficacy. Factors such as temperature, humidity, soil and leaf moisture, plant and pest growth stage, and edaphic conditions must be considered for best results, mainly when dealing with micro-organisms [82].

Although market and regulatory factors and pest resistance to traditional pesticides are driving the implementation of biological approaches, they still make up a small portion of the global crop protection portfolio. The obstacles that hinder the adoption of IPM techniques and the transition to organic agriculture are also barriers to the greater adoption of biological approaches. However, by increasing awareness and understanding of the benefits and histories of organic and IPM and identifying shared goals and priorities among their proponents and practitioners, we can enhance our effectiveness in overcoming these barriers and accelerate the adoption of biological approaches [89].

Strategies for speeding up the adoption of biological control options include increasing education and extension on proven, ready-to-use biological control alternatives, accounting for the total cost and benefits of biologically based alternatives to chemical controls, as well as implementing public and private policies that encourage biological control and reduce reliance on chemical controls. Collaborating on shared interests and goals can benefit both the organic and IPM communities of practice. Acceptable management practices include cultural practices and physical and mechanical techniques. The negative impacts of pesticides and their increasing ineffectiveness drive farmers to consider alternative methods to protect their crops. Farmers can adopt biological and cultural practices, reduce pesticide applications, and use reduced-risk pesticides. Finally, a systemic approach based on ecological principles can be implemented [89].

### 4.2. Cultural Practices

Agricultural practices that preserve a more prosperous community of biocontrol agents are essential to establish more vigorous and balanced agroecosystems that are less susceptible to pest outbreaks [90]. Creating more resilient agroecosystems requires a holistic approach integrating various techniques and practices. Landscape architecture (LA) is one such approach that can help improve agriculture. Initially, the concept of LA was focused on the aesthetic aspect of rural landscapes while incorporating science-based agricultural practices to ensure maximum food production. Later, integrated landscape management emerged, combining food production with conserving ecosystem services, particularly those provided by habitat biodiversity [87].

Encouraging grower compliance with best agricultural practices is crucial, given the demonstrated potential savings and the sustainable environmental profile of IPM [89]. Cultural practices reduce pest establishment, reproduction, dispersal, and survival. Changing irrigation practices can help, as overwatering can increase root disease and weed growth [90]. Selecting appropriate sites for cultivating field crops and fruit trees can help prevent infestations from insect pests in the future. The crops should be selected based on their suitability for the growing area and their tolerance to the prevalent pests and diseases in the region [90].

Mechanical and physical controls are methods used to control pests that involve directly killing, blocking, or making the environment unsuitable for them. They affect agricultural practices such as soil preparation, slash and burn, and manual weeding [91]. Soil steam sterilization is also effective for disease management, and barriers such as screens can help keep birds or insects out. Preventive practices are crucial to an IPM program [91]. These practices involve various methods, such as cleaning farm equipment (tillage equipment, haying equipment, etc.), planting certified seeds, and isolating infested crops or farmlands. These methods help prevent pests from being introduced into the field [91].

According to Baker, ensuring the effectiveness of biological or cultural practices on an ecosystem-wide basis often requires collective action [92]. The use of chemical control methods has been found to result in the loss of biodiversity. Herbicide drift can negatively impact non-target plants and arthropods, as highlighted by Egan et al. [93]. This can lead producers to invest less in biological control methods and rely more on chemical control methods. Safety problems and ecological disruptions continue to ensue, and there are currently calls for alternatives that are effective, safe, and economically viable [94].

Synthetic chemical pesticides are the most widely used pest control method; however, they present problems such as toxic residues, pest resistance, secondary pests, and pest resurgence [95]. Natural and organophosphate pesticides, are a more sustainable solution from an environmental perspective and have been proven effective in improving crop yields and plant biomass. Synthetic pesticides should only be used as a last resort or when necessary, and even then, they should be used only at specific times in a pest’s life cycle.

Natural control involves naturally occurring pest management methods to combat pests, such as using beneficial insects and diseases. Insecticides are used only when they are economically feasible and when the natural enemies cannot control the pests effectively [91,96].

Pesticides based on substances produced by microbes, like spinosad or microorganisms registered as plant protection products, are typically very selective. This means that they only affect a very narrow group of pests and do not harm other groups of organisms, such as beneficial insects [82,96].

Microorganisms and other biological substances are not entirely side-effect-free [97]. However, those registered as plant protection products undergo rigorous safety studies during the registration process to ensure their safety and lack of unwanted side effects. They usually decompose quickly and are less persistent than chemical pesticides [77,98].

If the total costs and benefits of biological, cultural, and chemical controls can be estimated, producers who adopt biological control can be compensated, reducing reliance on pesticides [89].

In agricultural development, pesticides have become an essential tool and plant protection agent to boost food production. Furthermore, pesticides play a significant role in maintaining many terrible diseases. However, pesticide exposure, both occupationally and environmentally, causes various human health problems. Studies have linked pesticide exposure to immune suppression, hormonal disruptions, decreased intelligence, reproductive abnormalities, and cancer [99]. The presence of pesticide residues in different crops has hurt the export of agricultural products [99]. To minimize human pesticide exposure, some main strategies include ensuring pesticide safety, regulating pesticide use, adopting appropriate application technologies, and implementing IPM [99]. More research is necessary to explore biologically based controls in conjunction with cultural practices and insecticides [100].

In the long term, IPM has additional sociological benefits, including improved employment opportunities, better public health, and enhanced well-being for agriculture workers. Despite the numerous advantages of IPM, stated so far, there are also some disadvantages. Not using pesticides regularly requires advanced planning and, therefore, a greater degree of management, such as selecting crop varieties resistant or tolerant to damage caused by pests and choosing cultivation systems that suppress damage predicted by pests while providing the crop with the highest yield potential. Another factor that can complicate IPM planning is climate change. To be a good IPM planner, it is vital to have a backup plan in case any problems arise. This will allow it to handle unexpected situations and maintain the effectiveness of this IPM strategy.

## 5. Effects of Light on Fruit Development, Color, and Flavor, and Strategies to Optimize Light Conditions through Pruning, Thinning, and Proper Orchard Layout

Light is a crucial factor in fruit growth and development, influencing various aspects such as intensity, duration, and quality. The interplay of these factors regulates physiological processes within the plant, ultimately shaping fruit development, quality, and yield. Previous studies have underscored the significance of light exposure in determining the yield and/or quality of several fruits, including grapes [101], apples [102], strawberries [103,104], tomatoes [105], sweet cherries [106], or blueberries [107]. 

Light serves as an energy source for numerous physiological processes, including photosynthesis [108,109] and chlorophyll biosynthesis [110], thereby impacting plant growth and development [108]. Insufficient light reduces photosynthesis and premature senescence in leaves [111], hampering plant growth and affecting fruit development, production, and quality [112]. Therefore, enhancing the photosynthetic rate of the plant is crucial for improving carbohydrate transport to the fruit [113], thereby influencing fruit yield [114] and quality [115,116]. Adequate light is also essential for pollinator activity [117], directly influencing flower and fruit yield [118] as observed by Cao et al. [118], who demonstrated that plants in sunny areas exhibited a larger floral display and received more pollinator visits per flower compared to those in shaded areas, resulting in improved fruit and seed production per flower.

Moreover, light influences the biosynthesis and accumulation of primary metabolites such as sugars, starches, and organic acids, vital for fruit growth, maturation, and quality [119]. These primary metabolites also impact the taste and flavor of the fruit [120]. Compounds like vitamin C directly rely on light availability [121]. Furthermore, secondary metabolites, including carotenoids, phenolic acids, and anthocyanins [122], are affected by light availability. Generally, secondary metabolites enhance fruit quality by contributing to fruit aroma, color, and nutritional value [123], aligning with consumer demand for high-quality fruits. Secondary metabolites also protect fruits against environmental stresses, particularly during postharvest storage [123]. 

Light also regulates plant hormones [124], such as auxins, cytokinins, gibberellins, abscisic acid, and ethylene. These hormones control various aspects of fruit development and ripening [125]. Auxins and gibberellins initiate fruit growth [126], while abscisic acid and ethylene play critical roles in fruit maturation and ripening [127]. 

### 5.1. Effects of Light Intensity, Duration, and Quality on Fruit Growth and Maturation

Light intensity, measured as the amount of available energy that reaches a surface area, influences the rate of photosynthesis and the overall growth of the plant, ultimately affecting fruit yield and quality [128]. Song et al. [129] concluded that an improvement in the photosynthetic capacity of tomato plants was positively correlated with an increase in light availability. 

Light intensity significantly affects the accumulation of carbohydrate substances such as sucrose, fructose, glucose, and starches, which are essential for fruit development [116]. The biosynthesis of anthocyanins, responsible for the formation of color in fruits, is also dependent on light exposure [130,131]. The works of Serrano et al. in sweet cherry [132], Zhang et al. [101] in grapes, and Peavey et al. [130] in pear are good examples of the correlation between light exposure and anthocyanin accumulation. Tombesi et al. [133] demonstrated that high light intensity reaching the kiwifruit surface at harvest produces high chlorophyll content associated with long shelf life during storage. Moreover, maintaining adequate light levels during postharvest storage is beneficial for preserving the nutritional quality of fruit [134] and its storage life. 

When light intensities exceed a critical level, photoinhibition can occur, creating unfavorable plant growth conditions that affect fruit growth [135]. Exposure of fruit to excessive sunlight can cause sunscald or sunburn [136], characterized by browning and necrosis. This can lead to yield loss and quality degradation of fruits [137]. Alternatively, low light intensities cause a reduction in plant biomass due to a decrease in photosynthetic rate [138] but also cause yield loss and quality degradation [139]. Kishore et al. [140] found that the fruit quality of pineapple deteriorates at low light intensity. Similar results were described by Pardossi et al. [141] in melon.

The adverse effects of sunburn on fruit crops such as grapes and apples are discussed in detail in Gambetta et al. [142] and Severino et al. [143], respectively. However, these responses may vary with species and/or cultivar [144], but also with the season of the year, region [143], and orchard management practices [142]. Under controlled conditions, particularly in greenhouse horticulture and growth chambers, low light intensity is one of the leading causes of reduced yield and fruit quality [145].

In addition to light intensity, the duration of light exposure throughout the day has a significant effect on several aspects of plant growth, namely, flowering [146] and fruit set [147], either directly or by inducing the endogenous circadian clock [148]. Circadian synchronization between chlorophyll production, CO_2_ fixation, and photosynthesis in the external period can enhance crops’ vegetative growth and productivity [149]. Optimal photoperiods vary among different fruit species and cultivars. This has been demonstrated by Sønsteby and Heide [150] in raspberry and by Garica and Kubota [151] in strawberry. Deviations from the natural light-dark cycle can disrupt developmental patterns and compromise fruit quality. For example, prolonged periods of darkness can delay fruit ripening [152] or induce physiological disorders [153], resulting in photoperiodic stress reactions [154]. This highlights the importance of maintaining an appropriate light–night cycle. In horticultural crops, additional lighting can artificially induce this cycle at night. For example, light supplementation in strawberries increased fruit weight, number of fruits, marketable yield, and fruit soluble solid content, but the results depended on plant genotype [155]. In apples, the extension of the light period induced the synthesis of anthocyanins in the peels and the accumulation of soluble sugars in the flesh, improving the fruit quality [156]. In tomatoes, Jiang et al. [157] and Paponov et al. [116] found that light supplementation positively affected fruit size and chemical characteristics. The time of light supplementation also affects the fruit characteristics. For example, in tomatoes, morning light supplementation improved the nutritional quality of tomato fruit, while evening light supplementation improved their flavor [105].

The quality of light, characterized by its wavelength composition, also has distinct effects on fruit development and maturation [107,158]. Among the different light wavelengths, blue and red light play critical roles in plant growth and metabolism [158] and in yield and fruit quality [105]. Fruit quality is also affected by white light and yellow light [159]. Blue light affects the accumulation of anthocyanins in fruits, while red light induces its biosynthesis [158]. The balance between blue and red light affects fruit characteristics such as color, flavor, and nutritional content. In watermelon, the red exposure maintained the original color of the fruit and redness and delayed aroma deterioration [135]. Conversely, in grapes, the highest color intensity, sweetness, and skin edibility sensory scores were achieved under a combination of blue and red light [101]. This means differences in response to light wavelengths strongly depend on the species and specific spectral wavelengths applied.

Changing the spectral composition using artificial lighting technologies, such as light-emitting diodes (LEDs), allows altering fruit traits in terms of aesthetic, nutraceutical, and nutritional properties [160] to meet market demands and consumer preferences.

### 5.2. Management Practices to Optimize Light Conditions

Light management in fruit orchards is crucial to maximizing yields of good quality fruit that meets market and consumer demands. Through strategic pruning, thinning, and thoughtful orchard design, producers can effectively manage the distribution and intensity of light to benefit fruit production in natural and greenhouse environments. 

Pruning is a fundamental orchard management practice to improve canopy access to light, ensuring the right balance between vegetative and reproductive growth required for consistent, high-quality yields [161]. The highest yields result from high light interception and distribution, with minimal shading within the tree [162]. Removing excess and unproductive branches, leaves, and flower buds can increase light penetration into the canopy and improve air circulation, reducing disease pressure [162,163] and promoting fruit size, yield, and quality [164]. However, it is vital to avoid severe pruning, as this can contribute to removing more photosynthetic biomass, which may result in lower yields and fruit quality [162].

Pruning significantly affects fruit yield, marketability, and various indicators of fruit quality. The positive effect of pruning on yield has been described in several fruit crops, including cocoa [162] and banana cv. Grande Naine [165], apple [166], blueberry [164], peach [167], orange cv. Valencia [168], and citrus [163]. Therefore, the ability of pruning to increase yield depends on several factors, including training system [162,167], pruning technique [161], pruning severity [162,168], and planting density [161]. Pruning also positively affects several aspects of fruit quality. Al-Saif et al. [168] found that different pruning intensities performed on “Valencia” orange trees positively affected fruit quality traits such as size, firmness, juice content, total soluble solids (TSS), TSS/acid ratio, and vitamin C content. Singh [169] also reported improved fruit color in peaches, making them more visually appealing.

In addition to pruning, the removal of excess fruit clusters or individual fruits by thinning is an essential orchard practice to improve light penetration to the lower canopy [102], optimize resource allocation, and prevent heavy fruit loads and/or alternate bearing [170], which are common in fruit species. Thinning allows the remaining fruit to receive adequate sunlight, water, and nutrients, resulting in maximum commercial yields and fruit of good market size [171]. In terms of fruit quality attributes, thinning improves fruit taste [172], color [173,174], and nutrient contents, including sugars [175], anthocyanins [175,176], and total phenolics [177], but the effects depend on the thinning intensity [178], species, and/or cultivar [177]. Fruit thinning is particularly important in stone fruits [179] but also in apples [180], pears [181], grapes [175], and oranges [171], among others.

Manual, mechanical, and chemical methods can achieve fruit thinning. Among them, manual fruit removal is expensive and labor-intensive [171,182,183], and despite its selectivity, it is challenging to improve return flowering [184]. Mechanical and chemical fruit thinning can be alternatives to manual thinning [185]. The advantages and conditions of using mechanical thinning have recently been reviewed by Lei et al. [183]. Chemical thinning, although cheaper than the other approaches, is highly dependent on weather conditions [184], cultivar, chemical formulations, and their concentration [186,187]. In addition, it can impact the environment and cause phytotoxicity [182]. Therefore, there is a growing demand for more environmentally sustainable solutions that have proven effective in improving crop yields and plant biomass. 

Orchard design is crucial in optimizing light conditions, particularly in greenhouse environments with limited availability. Minimal or excessive levels of light interception can inhibit optimal production [161], so light interception should be optimized rather than maximized. The ultimate goal of orchard design is to ensure consistent, high yields of high-quality fruit [102]. The strategic placement of trees and the correct choice of tree spacing (distance between rows and between individual trees) will allow light to penetrate the canopy. A north-south row orientation is generally preferred as it maximizes sun exposure on both sides of the canopy throughout the day [188], minimizes shading [189], and ensures balanced fruit development [190]. In this regard, the early work by Mota et al. [191] showed that north-south Syrah winter wines had higher levels of color intensity, anthocyanins, total phenolics, and total phenolic index, ashes, and pH than east-west vines. Light interception can be increased by suitable tree shape and size [192] and density planting [189]. Reduced spacing within and between rows promotes better productivity. Still, it can lead to increased shading and uneven distribution of light across the tree canopy, reducing the fruit’s nutritional and pharmaceutical value [102]. In turn, increased tree spacing reduces competition for light, increasing the yield quality and, thus, market value [193]. Therefore, selecting the appropriate training system is crucial as it controls the tree’s shape and height and promotes light interception (see Section 8 for more details).

## 6. Importance of Pollination on Fruit Set and Quality and Strategies to Enhance Pollination

Pollination is a crucial feature of many crops [194]. It can be defined as the delivery of pollen from anthers to stigmas by vectors, both biotic or abiotic, and including the stages: (i) removal of pollen from the anthers, (ii) transportation of pollen, and (iii) deposition on a receptive flower [195]. Several works show that pollination is essential for agricultural production, with 1500 crops requiring insect pollination and 70% of the crops directly used by humans depending on pollinators [196]. Limitations in pollen availability can result from several factors, namely, the absence of pollinators, lack of synchrony between plants and pollinators, insufficient pollen, or asynchrony between pollen production and stigma receptivity [197]. However, pollination is still among the least understood factors when considering orchard management [198]. Understanding pollination needs within the orchard is critical to determine whether this service is optimal or can be improved. The intensification of agriculture in the last decades has resulted in a decline in bees and other pollinating insects [199], linked to land-use changes, widespread use of insecticides, herbicides, and fertilizers (indirectly affecting pollinators by decreasing the availability of flowers) [200]. The use of fungicides, once considered benign for bees, has now been shown to hurt wild and managed pollinators through direct toxicity or synergism with insecticides [201]. Other factors, like the dissemination of non-native species and diseases, as well as climate changes, are other challenges pollinators face [202,203].

The foreseen climate change will affect both the life cycle of the plants and pollinators. In plants, changes due to rising temperature have been noticed, resulting in early flowering, while pollinators increase their overwintering, geographical distribution, non-native species, and changes in growth rates [204]. Models have shown that climate change will reduce the probable occurrence of pollinators across all crops by 0.13% by 2050 [205]. Climate change might also result in variations in ecological responses, like species distribution ranges, phenological dislocations, new interactions between previously isolated species, extinctions, and other unforeseen effects [206,207,208,209,210].

Several studies show the losses in crop production in the absence of pollination services [211,212], but these services are still not considered when economical, agronomic management and agronomical policies are defined [213]. A key factor is maintaining agricultural heterogeneity to preserve pollination services [214]. Still, the current trend is the cultivation of pollinator-dependent crops linked to decreased agricultural diversity, leading to reduced productivity [215]. Continuing this tendency, pollination services [210] have severe consequences, such as disruption of plant-pollinator networks [216], with all the associated impacts. Limitations of pollination have been recorded in several crops, including macadamia [217], shea [218], cacao [219], apple [220], custard apple [221], or eggplant [222]. These shortages in pollination may endanger the global food system, precisely crop production and trade [223,224]. However, measurement of crop dependence on pollinators, determining which pollinator taxa contributes the most to the target crop, or even how a given cultivar relies on pollinators [225], and managing and conserving pollinators and pollination services are challenging to achieve [194]. Nonetheless, management practices can and should be used to increase crop pollination [226]. Practices implemented to promote wild or managed pollinators have been shown to improve crop pollination [227]. For instance, increasing the richness of flowering plants by adding flower strips or margins or reducing pesticide use has been shown to stimulate the diversification of wild pollinators, thus providing additional pollination [228,229,230]. The most commonly managed pollinator species is the honeybee (*Apis mellifera*). However, other bees, such as species of the genus *Bombus* spp. and solitary bees of the genus *Osmia* spp., can also be employed to promote pollination of crops [231,232,233]. Under certain limited conditions, such as the impossibility of improving natural pollination or when pollen quantity is reduced, economic and commercial goals of crops might only be achieved by artificial pollination. This approach involves high costs, labor, and mechanical force but has to be used in certain crops (like kiwifruit [234], pistachio [235], almond [236], apple, cherry, walnut [237], olive tree [234,237], or hazelnut [238].

The link between crop quality and pollination has also been found for some crops. Indeed, adequate pollination can enhance quality in entomophilous crops, namely, orchard fruit production (e.g., in apple [239]), but also in field crops (oilseed rape [240]) and small fruits and vegetables (e.g., strawberry [241]; tomato [242]; bell peppers [243]; highbush blueberry [244]). Other studies point out different effects of pollination on quality and yield traits. For instance, Bartomeus et al. [245] indicate an enhanced yield ranging from 18% in oilseed rape to 71% in buckwheat. Besides yield, quality changes were also linked to pollination, with commercial-grade strawberries increasing, as did the oil content in oilseed rape. A meta-analysis has also shown an increase in the overall quality of crops by 23% when comparing the effect of pollination service on open-pollinated and pollinator-excluded plants [246]. The most significant effects are recorded in organoleptic traits rather than nutritional ones, with all pollinator groups equally affecting the quality of fruits and vegetables. These positive effects on quality by pollinators are most likely linked to the activation of phytohormonal processes due to successful fertilization [247], namely, the auxin-mediated promotion of gibberellins, which regulates seed formation and fruit growth [248]. These same phytohormones increase pulp firmness, helping to enhance the post-harvest quality of fruits [249,250]. While the effects on organoleptic traits are significant in fruits and vegetables, reduced impacts on micro- and macronutrients are likely caused by a dilution effect created by the rapid cell expansion and increased water content as fully pollinated fruits grow in size [251,252]. Furthermore, the effect of pollination on nutritional characteristics can also be linked to specific synthetic processes during fruit formation and to the high instability that such compounds present during fruit metabolic activity [251]. Thus, fruit size and appearance seem more reliant on pollination than other yield-enhancing factors rather than fruit nutrients.

### Weeds, Hedgerows, Managed Pollinators, and Pollinators Services

The decline of natural pollinators is linked to lower flower availability due to isolation from critical floral and nesting resources [253,254,255], which results from promoting monoculture agriculture [256,257]. Flower diversity generates resource diversity in more heterogeneous landscapes, providing stable food for bees throughout the season [258]. Although some pollinators can benefit from the condensed flowering time within one monoculture, the overall effect on the pollinators’ community is poorly understood from a long-term perspective [259]. Floral diversity also enhances the population of polylectic species [260,261] and ensures the stability of pollinator densities [262] and the diversity of pollinator communities [263,264,265]. Studies reaffirm that maintaining the food web to create a higher diversity of pollinators is critical, even mitigating the known loss in honeybee populations [263,266,267]. Hence, it is vitally important to enhance the pollinator habitat, with the most common strategies being field margin manipulation, including non-crop buffer strips to provide nesting locations, restoration of native plants in adjacent areas, and wildflower strips sown with pollen- and nectar-rich plants [268,269]. Current agricultural practices that result in a homogenous landscape do not allow room for weeds, which could offer many necessary conditions for beneficial insects, namely, pollen nectar or microhabitats [270]. This adds value to the non-crop forage plants (often regarded as weeds), as they provide alternative food resources (pollen/nectar, alternate prey/host), thus aiding in the survival of viable populations of pollinators before, during, or after crop bloom [271]. Whenever weeds are not allowed, the alternative is the use of hedgerows and weedy plants such as nettles, wild *apiaceae*, comfrey, and wild clovers, but also herbaceous plants, namely, long-corolla perennials that usually have more nectar than annual plants [272]. The hedgerows will, as weeds, provide additional pollen and nectar for adults and food plants for larvae, as well as shelter and nesting sites [273]. Hedgerows may also be attractive because they provide sheltered habitats for several native and woodland-adapted edge plant species [274,275]. Seed mixtures containing many “grass and wildflower” or “nectar and pollen” seeds (containing plants considered weeds) are commercially available, and they can be sown around agricultural plots, attracting pollinators [276] and containing specific host plants for larval stages of butterflies, moths, and beetles. Even so, and despite the increasing popularity of field-edge plantings, studies have recorded mixed results. Increases in wild bee abundance, richness, and crop yields have been recorded [277,278,279], while other authors recorded non-significant effects [280,281]. However, information on how crop type and landscape influence and the evaluation of crop response (fruit quality or yield) is still lacking, and it is essential to elucidate their effect on crop pollination and pollinator conservation.

As mentioned, crop pollination is often overlooked when considering crop management, which is provided almost entirely as an ecosystem service. Even so, managed pollinators are used in some crops, including honeybees, which have long been used to ensure pollination, reducing reliance on wild pollinators [196,282]. The amount of managed hives grow worldwide globally [283], but this might not relate to less restraint in agricultural production due to pollen limitation, mainly for three reasons: (i) the need for agricultural pollination services grows faster than supply [283]; (ii) the number of honeybees has increased unequally in countries, with higher growth in honey-producing countries, but declines in the remaining countries; (iii) in most countries [282], honeybees are primarily used as honey producers, with pollination of crops being an ecosystem service rather than an intentional input. More importantly, pollination shortage (i.e., limitation of yield by incomplete pollination) could limit the yield of crops highly dependent on pollinators even in the absence of any of the so-called “pollination crisis” (i.e., pollen temporally increased limitation due to recent decreases in biotic pollination). Indeed, as highly productive crops tend to flower intensively for a short-term period [284,285], pollinator communities may not satisfy requirements for fertilization, and pollination shortage can occur even in non-degraded pollinator communities and natural ecosystems [286]. However, relying on honeybees as a managed pollinator has some drawbacks that must be considered. For instance, honeybee hives can reduce limitations in pollen availability in self-compatible crops but not in self-incompatible crops [287]. Furthermore, even considering the generalist characteristics of honeybees, several crops are not efficiently pollinated by them [217,288], and, finally, there is an increase in overwintering losses of bee colonies [289,290]. Therefore, using the already referred alternative managed pollinators like bumblebees (*Bombus* spp.) and solitary bees like *Osmia* spp. or *Megachile* spp. is interesting. Even so, data indicate that the presence of honeybees, even if not entirely needed to meet pollination needs, can help create a synergistic interaction between species, increasing the efficiency of wild pollinators through behavior modification [291].

## 7. Advancements in Genetic Engineering and Molecular Techniques for Selecting and Breeding Varieties with Desired Traits

Crop improvement aims to increase crop yield and quality. Recent advancements in agricultural technology have led to increased productivity, but there is now a focus on customer preferences for higher-quality produce. Consumers want fruits with appealing appearance, texture, flavor, aroma, and high nutritional value. Various methods have been used to improve crop attributes, including conventional breeding, mutation breeding, molecular-marker-assisted breeding, and genetic engineering. Genetically modified crops have succeeded, but concerns persist regarding health and environmental impacts [292].

Advancements in genetic engineering and molecular techniques have revolutionized the field of plant breeding, enabling precise and efficient selection of varieties with desired traits. In Table 2, we present some key advancements: (i) CRISPR-Cas9 (clustered regularly interspaced short palindromic repeats) and Genome Editing have revolutionized genome editing, enabling targeted modifications in the DNA of plants with high precision. Continuous improvements in CRISPR techniques, such as base editing and prime editing, have increased the efficiency and accuracy of genetic modifications. This allows for introducing specific traits without introducing foreign genes [293,294]. (ii) CRISPR-Cas Technologies Beyond Cas9, wherein other CRISPR-Cas systems, such as Cas12, Cas13, and Cas 14, have been explored for their potential in genome editing and gene regulation. These systems offer additional capabilities, including targeting RNA (Cas13) for gene regulation and providing alternative options for precise genetic modifications [295]. (iii) High-throughput sequencing and omics technologies allow for the comprehensive analysis of genomes, transcriptomes, and proteomes on a large scale, having become more cost-effective and accessible, facilitating the rapid identification of genes associated with desired traits. This information aids in the selection of superior breeding candidates [296]. (iv) Marker-assisted selection (MAS) involves using molecular markers linked to specific genes or traits for selection in breeding programs. The development of high-density marker platforms and advanced statistical methods has improved the efficiency of MAS, allowing for more accurate and rapid identification of plants carrying desired traits [297]. (v) Genomic selection involves predicting the performance of plants based on their entire genome rather than individual markers. Advances in computational biology and machine learning have enhanced the accuracy of genomic prediction models. This approach benefits complex traits influenced by multiple genes [298]. (vi) Synthetic biology and gene synthesis, involving designing and constructing new biological parts and devices, have become more efficient and cost-effective, creating synthetic genes or pathways. This allows for the engineering of crops with novel traits or the optimization of existing pathways [299]. (vii) RNA interference (RNAi) and gene silencing, a technique used to silence or downregulate specific genes by introducing small RNA molecules. Improved delivery methods and enhanced understanding of RNAi mechanisms have made it a valuable tool for targeted gene silencing, particularly for controlling pests and diseases [300]. (viii) Gene stacking and multigenic approaches involve combining multiple genes or traits into a single crop variety, allowing breeders to develop varieties with resistance to numerous pests or diseases and tolerance to various environmental stresses [301].

These advancements collectively contribute to developing crops with improved traits, addressing challenges such as increased yields, resistance to pests and diseases, enhanced tolerance to environmental stresses, and pleasant sensory characteristics. As technology continues to evolve, the future promises even more sophisticated and targeted approaches to plant breeding.

### 7.1. Important Endonucleases for Gene Editing Aimed to Improve Crops’ Sensory Traits and Consumer Acceptance

Certain enzymes called sequence-specific nucleases (SSNs) can introduce mutations at a specific location by adding, deleting, or altering the DNA sequence [302]. These SSNs are categorized into three groups, namely, CRISPR/Cas9, ZFNs, and TALENs, and are widely used for genome editing [303].

ZFNs, TALENs, and CRISPR-genome-edited plants may be considered non-GMO (genetically modified organisms) under existing EU regulations [304]. This could change European perceptions of GMO biotechnology and open up commercial opportunities for fruit crop improvement.

#### 7.1.1. CRISPR/Cas9 Technology of Gene Editing

The CRISPR/Cas9 system is a revolutionary genome editing technology that has gained significant attention due to its high efficiency and adaptability in manipulating the genes of various organisms. Derived from the adaptive immune systems of bacteria and archaea [305], the CRISPR method, originally a bacteria’s defense mechanism against viral infections, has proven effective in editing plant genomes [292,306].

The CRISPR/Cas9 system comprises two major components: guide RNA (gRNA) and the Cas9 nuclease (Figure 2). Unlike ZFNs (zinc finger nucleases) and TALENS (transcription-activator-like effector nucleases), which rely on protein–DNA binding for target sequence specificity, the CRISPR system involves RNA–DNA binding. The Cas9 nuclease has a recognition domain consisting of two RNA-binding domains and a protospacer-adjacent motif (PAM) domain, which helps it bind to the target DNA. The nuclease domain of Cas9 cleaves the DNA at the target location with the help of the HNH and RuvC-like nuclease domains within the Cas9 protein, resulting in double-strand breaks (DSBs) at the target DNA sequence [292,307] (Figure 2).

The gRNA is a synthetic RNA that guides the Cas9 nuclease to the target site. It consists of two components: CRISPR RNA (crRNA) and transactivating CRISPR RNA (tracrRNA). crRNA provides information on the complementary sequence of the target DNA, whereas tracrRNA combines with crRNA to form a complex that assists in the assembly and stabilization of Cas9-gRNA. The gRNA complex guides the Cas9 nuclease to the target site, inducing DSBs at the gRNA complementary site sequence. Following the cleavage, DNA repair mechanisms are triggered, and the DSBs are repaired either via homology-directed repair (HDR), which uses the template DNA to repair the DSBs, or via non-homologous end joining (NHEj), which produces insertions or deletions (indels) that can disrupt the genes [292,307].

Grape berries are rich in nutrients that are beneficial to human health. However, conventional breeding is a time-consuming process for grapevine improvement. CRISPR/Cas9 technology is a promising approach for crop improvement and gene functional study in grapevine. Recently, it has been widely used for trait improvement and gene study [308,309,310,311,312,313,314]. Many of these studies focus on improving grape varieties’ resistance to pathogens [294,313] or enhancing intrinsic water use and efficiency, indicating potential advantages in reducing stomatal density under future environmental drier scenarios [314].

Gene editing using the CRISPR/Cas 9 technique has also been applied in fruit ripening, improving fruits’ sensory traits [315]. Tomatoes are one of the most studied fruits regarding fruit ripening, texture, and color variations. For example, mutations in the ripening inhibitor (*RIN*) gene can prevent tomato fruits from ripening fully [316]. In a study by Ito et al. [317], researchers used the CRISPR/Cas9 system to knock out the *RIN* gene in tomatoes. They then analyzed the resulting *RIN* mutant and found that the mutation did not affect the ripening initiation and exhibited moderate red coloring. This finding suggests that the *RIN* gene is unnecessary to initiate fruit ripening.

Osorio et al. [318] have reported that tomato breeders use *RIN* mutation to develop better hybrids, but it negatively impacts the flavor of the tomato fruit. Li et al. [319] have pointed out that using CRISPR/Cas9 technology to obtain *RIN*-deficient fruits can reduce ethylene production and affect the synthesis of volatile substances, carotenoids, and the fruit ripening process. Low ethylene production in *RIN*-deficient fruits prevents the induction of ethylene production in the autocatalytic system-2. They also lack volatiles, carotenoids, and transcripts related to these pathways. Li et al. [319] also supported the idea that fruit ripening requires the participation of ethylene response factors (ERFs), *RIN*, and ethylene. Ethylene initiates the maturation of green fruit and affects the expression of *RIN* and other factors, completing the entire ripening process of fruits [319].

Another important sensory aspect highly appreciated by consumers is fruit texture. Changes in fruit texture during transportation and storage can lead to significant decay and the development of typical post-harvest diseases. These changes are caused by the alteration in enzyme activity, affecting the structure of cell walls. Therefore, fruit texture is crucial in determining fruit quality and shelf life [320,321]. CRISPR/Cas9-induced mutation of the tomato pectate lyases (*PL*) gene can increase fruit’s firmness and prolong shelf life without adversely affecting other aspects of fruit ripening [322].

Yu et al. [323] improved tomato texture by working with *alc* mutation, which has a single base pair mutation in the NOR gene. This mutation changes the amino acid sequence and has been found to not only extend the shelf life of the fruit but also make it more flavorful and resistant to disease. The CRISPR/Cas9 method achieved the mutation through the HDR recombination pathway. Without T-DNA insertion, the resulting homozygous *alc* mutant has improved the fruit’s storage time and increased shelf life.

It is widely known that fruit color is determined by the pigments present in it. Genes responsible for pigment synthesis influence the bioactive compounds and affect the fruit color, which plays a vital role in consumer choice. For instance, Europeans and Americans prefer red tomatoes, while Asians prefer pink ones [324]. Research on *SlMYB12* has shown that it affects flavonoid accumulation, and mutation of this gene can result in pink-colored tomato fruits [325]. Moreover, the CRISPR/Cas9 system can produce purple tomato fruits by enhancing anthocyanin accumulation. This can be achieved through the ant1 mutation [326]. Additionally, tomato pectin-degrading enzymes such as PL, polygalacturonase 2a (PG2a), and β-galactanase (TBG4) affect fruit ripening. Wang et al. [327] obtained silent mutants of *pl*, *pg2a*, and *tbg4*, and, interestingly, *pg2a* and *tbg4* CRISPR strains did not soften the fruits but affected their color.

Besides tomatoes, other crops have also been tested and improved using CRISPR/Cas gene-editing technology, namely, bananas, where Kaur et al. [328] performed a study where complete albino and variegated phenotypes were observed among regenerated plantlets; apples and grapevine, where higher acidic fruits were obtained by increasing tartaric biosynthesis [329]; grapes, with the production of albino phenotypes [330]; groundcherry, aiming to improve the fruit size [331]; kiwifruit, seeking to obtain a faster fruit development [332]; and watermelon. According to Ren et al. [333], knocking out genes responsible for sugar transport, such as ClAGA2, ClSWEET3, and ClTST2, can impact sugar accumulation in watermelon. By analyzing the quantitative trait locus (QTL) that affects the levels of sugars, lycopene, β-carotene, citrulline, and arginine biosynthesis, we can target these regions for gene editing using CRISPR/Cas9 technology. This can create genetic variation and help in selecting and advancing new progenies.

Recently, editing of the sucrose transporter gene *ClVST1* using the CRISPR/Cas9 gene editing platform resulted in the development of watermelon lines with decreased sugar content. In contrast, overexpression of *ClVST1* increased sucrose content in the mutant lines [334].

#### 7.1.2. ZFN Technology of Gene Editing

Zinc finger nucleases (ZFNs) are hybrid proteins comprised of engineered endonucleases and artificial fusion proteins that connect a zinc-finger DNA-binding domain to a non-specific DNA cleavage domain of the FokI restriction endonuclease. An engineered ZFN consists of ZFN monomers linked to an 18–24 bp DNA sequence with a spacer. The ZFN domain identifies the target DNA sequence, whereas the FokI domain cleaves the DNA, inducing modifications [303,335].

Using ZFNs technology, the *LEAFY-COTYLEDON1-LIKE4* (*L1L4, NF-YB6*) gene of the nuclear transcription factor Y (*NF-Y*) transcription factor was disrupted in tomatoes, resulting in *L1L4* mutants. The metabolic contents of the mutants, such as oxalic acid, citric acid, fructose, β-carotene, total phenols, and antioxidants, were found to have slight variation compared to the wild types. It was concluded that *L1L4 TF* plays a vital role in regulating biosynthetic pathways of seed storage proteins and fatty acids in tomato fruit and seeds [336]. However, few works applied this technique to improve the sensory traits of fruits other than tomatoes.

#### 7.1.3. TALENs Technology for Gene Editing

TALENs, short for transcription activator-like effector nucleases, are endonucleases that can create double-stranded breaks (DSBs) in specific target DNA sequences. They are similar to ZFNs, a different type of endonuclease, in that they also have a DNA-binding domain derived from transcription activator-like effectors (TALEs) and a nuclease domain from the FokI endonuclease [337]. The DNA-binding domain of TALENs consists of multiple repeats of TALEs. Each repeat identifies a specific nucleotide in the target sequence [338]. On the other hand, the nuclease domain from the FokI endonuclease requires dimerization for DNA cleavage. TALENs are typically used in pairs that target one each of the DNA strands. Once they enter the host cell, the DNA-binding domain of the TALEN attaches to the target site, and the FokI domain dimerizes. This forms a functional nuclease complex that induces DSBs at the target site [303].

Lor et al. [339] reported using TALENs to target the *PROCERA* gene in tomatoes. They introduced the construct into tomato cotyledons via *Agrobacterium*-mediated transformation and recovered transgenic lines. From the regenerated lines, 15% contained mutated *PROCERA* alleles, and two selected lines resulted in heritable mutations. Homozygous progeny had an enhanced GA response.

Recent gene-editing technologies such as ZFNs, TALENs, and CRISPRs significantly impact plant biotechnology. These tools allow the targeted modification or mutation of specific genes related to sensory attributes and consumer preferences without introducing foreign DNA. As a result, the plants developed with this technology may be considered non-transgenic genetically modified plants. This opens up the possibility of growing fruit crops with superior phenotypes and making them commercially available, even in countries where genetically modified crops are not widely accepted. An example of such fruit crops is the genetically engineered, non-browning apple, which retains its natural flavor and taste and does not go brown after a fresh cut. The Arctic apple is another fruit engineered to resist browning after being cut. These apples are only available in the US and have received approval from the Food and Drug Administration. However, if they were approved in Europe, they would have to be labeled genetically modified. The manufacturers claim that the primary benefit of these apples is reducing food waste [340].

## 8. Modern Canopy Management and Innovative Training Systems to Improve Yield Efficiency and Fruit Quality

The canopy of a fruit tree comprises its physical elements, such as the stem, branches, shoots, and leaves. Canopy density is influenced by the quantity and size of the leaves, while factors like the number, length, and orientation of the stem, branches, and shoots influence canopy architecture. Canopy management in fruit trees involves developing and maintaining the structure concerning size, shape, and branch orientation. These aspects impact water transport, transpiration, carbon acquisition, and allocation to optimize productivity and fruit quality [341,342,343,344]. The fundamental concept in the canopy management of perennial trees is to efficiently utilize the available land and climatic conditions for increased productivity using three-dimensional approaches. Canopy management employs various techniques to adjust the position and quantity of leaves, shoots, and fruits in space, significantly influencing the plant’s geometric structure, including the spatial distribution of leaf area and leaf orientation. Canopy management represents an interrelation of the physiology underlying the relationship between vegetative growth and production. In fruit tree training, understanding fruit trees’ geometrical and topological characteristics proves beneficial for fine-tuning their shape. This, in turn, enhances fruit production in quantity, regularity, and quality [345].

Canopy architecture also plays a significant role in adapting trees to stress conditions. Dwarf cultivars with dense crowns demonstrate increased tolerance to drought by reducing dehydration compared to wide-crown varieties. The compactness of the canopy, achieved through reduced leaf size and a change in crown shape, helps lower transpiration rates. This compact structure decreases the penetration of solar radiation, thereby improving the microclimate within the tree. As a result, dwarf cultivars with compact crowns exhibit greater drought tolerance than open-crown varieties by delaying dehydration [346].

### 8.1. Canopy Management in Modern Agriculture

Canopy management in modern agriculture aims to improve tree health, productivity, and fruit quality [347]. It ensures adequate fruiting units, allows sufficient ventilation within the canopy, and prevents the overlapping of foliage to minimize the occurrence of parasitic leaves [348]. Moreover, it should provide adequate coverage for sprays and prevent the formation of a microclimate conducive to developing pests and diseases. Canopy management aims to attain higher yields with good quality; maintain a proper balance between root and shoot growth; establish strong crotches; eliminate unwanted, overcrowded, dead, diseased, and pest-affected shoots; regulate tree architecture; and achieve the desired shape for a high-density planting system [349]. The goal of canopy management encompasses facilitating practices such as spraying and harvesting, optimizing the use of ventilated and temperature, regulating the plant’s exposure to air, and ensuring accessibility to machinery between rows [350].

Cultivating smaller trees has increased production and improved fruit quality in numerous fruit crops. This method revitalizes declining productivity and fruit quality in large, overgrown orchards. Small trees outperform large ones in capturing and converting sunlight into fruit. It reduces additional expenses associated with harvesting large trees and mitigates safety risks for harvesters working with more giant trees [351].

Despite a lower level of innovation in variety development in pear orchards, a wide range of orchard design systems, from low to ultra-high density, is already established worldwide [352]. Generally, there has been an increased planting density in pear orchards facilitated by dwarfing rootstocks. In recent years, especially in Europe, there has been a gradual intensification due to quince accessions characterized by limited vigor [353,354]. The possibility of using rootstocks with different vigor and the plasticity of the pear allows the tree to be shaped in various forms, contributing to the high diversity of training systems adopted worldwide [355,356,357].

Citrus canopy management involves two essential components: selective pruning and effective control of resulting vegetative regrowth. Selective pruning requires precise timing based on the phenological cycle of the trees, employing accurate cutting techniques to remove specific unwanted branches. Regrowth management typically entails thinning out excessive vegetative shoots that emerge post-pruning. Canopy management is valuable for inducing precocity and sustaining high production of optimally sized, high-quality fruit. Additionally, it complements various other management practices, including interrow access, fertigation, maintenance of irrigation systems, scouting, pest and disease control, weed management, and grove sanitation and harvest activities [347].

### 8.2. Innovative Training Systems

Training involves tying, trimming, attaching, or arranging a plant to give it a specific form, regulating tree planting and canopy formation to optimize fruit yield and quality [358]. Over the years, various training systems have been adopted to increase production and maintain fruit quality. This effort has been made because an inappropriate training system can result in canopy shading, consequently reducing the qualitative parameters of the fruit, such as size, color, sugar content, and concentrations of secondary metabolites. Training systems influence branches’ development, position, and angle, consequently affecting both yield and fruit quality [359]. These systems simplify tree architecture and facilitate efficient use of orchard space. Additionally, a well-designed training system may contribute to regulating tree structure, promoting the formation of flower buds, and mitigating the adverse effects of shading on fruit development. The selection of training systems should also align with planting density and regional adaptation [360]. In this context, innovative training systems involving novel approaches to shape and support fruit trees have been developed, fostering enhanced growth and fruit development. These systems may include techniques like “Espalier,” “Trellising,” “Spindle,” “Thin vertical,” “inclined V,” “Palmette hedgerow,” “Pyramid hedgerow,” “Bi-axis,” or specific pruning methods tailored to the characteristics of the fruit tree [167,361,362]. Research on “V-shape” systems has primarily focused on apple and pear trees [363,364,365,366,367,368,369,370,371,372]. The most famous “V-shaped” canopy systems, recommended as an alternative for orchards with high tree densities, are the “Güttingen-V system,” the “Y-system,” the “Drilling,” and the “Mikado system” [373]. These systems promote a good yield of high-quality fruits [374,375,376]. The optimum angle from vertical for a leader to maximize the fruit size is about 60 degrees. In the case of fruit color, the best results are obtained with leaders growing vertically [377]. Numerous studies indicate that dense planting of spindle trees suppresses their vegetative growth. It enables high yields, but at the same time, it tends to compromise their quality in terms of fruit average weight, size, and coloration [358,362,378]. Other authors emphasize the high costs of wire supports and the necessity of laborious tree pruning and training as significant drawbacks of V-shape systems [377].

Various training systems have been experimented with, for example, in peaches (*Prunus persica* L.), each serving specific objectives, such as optimizing water utilization, improving labor efficiency, and designing canopy architectures compatible with mechanization and robotics. Generally, higher planting densities contribute to increased yields, but excessive densities may lead to shading issues, and an overwhelming crop load could negatively impact fruit quality. An ideal peach cropping system aims to balance maximizing yield potential and ensure the highest potential fruit quality. Effective management of high-density peach fruiting wall systems can result in improved and consistent fruit quality, contributing to a sustainable industry [379].

### 8.3. Technology to Mechanize Canopy Management

In wine grape production, the use of machines for essential tasks such as pruning and canopy management is increasing. These tasks are labor-intensive and time-sensitive, and wine grape production regions face rising wage costs and labor shortages [380]. Generally, in the fruit production process, the workforce accounts for 30–45% [381]. The proper execution of these practices significantly impacts fruit yield and quality. Mechanization offers advantages in terms of timeliness, uniformity, and cost-effectiveness. However, it is crucial to note that many existing methods are nonselective, and achieving optimal results demands meticulous attention to vineyard design, management, and machine settings [380].

The mechanical shoot thinner is non-selective, and thus it does not target specific shoots. Instead, it strikes the canopy at an adjustable frequency, reducing shoot density. When applied early, mechanical shoot-thinning decreases leaf layers, enhancing cluster exposure to moderate sunlight levels. This, in turn, improves the phenolic content of red grape varieties [382,383,384]. During the spring growth flush, shoots may develop from latent buds on the trunk. Traditionally, these unwanted shoots were removed manually. However, the use of machines for this practice has increased [385].

The goals of fruit-zone leaf removal may include limiting the crop level by reducing the number of berries set, improving cluster exposure to sunlight depending on timing, and enhancing airflow to reduce fungal infections. However, the outcome must balance the crop level with the exposed leaf area. Various types of equipment can mechanically remove leaves in the fruit zone [386]. Achieving a balance between crop level and exposed leaf area requires precise timing for fruit-zone leaf removal. If conducted before bloom, the grapevine tends to reduce the number of berries set, leading to greater exposure with reduced yield [387,388]. On the other hand, if conducted after bloom, the grapevine experiences increased solar radiation and temperature in the fruit zone [389]. The impact of fruit-zone leaf removal on yield can vary depending on the timing and climate of fruit growth. In warm regions, fruit-zone leaf removal often has a minimal effect on yield. However, an improvement in the leaf area-to-fruit ratio can be expected due to removing excessive leaf layers that shade the cluster in the fruiting zone [390,391,392].

In fruit trees, the time-of-flight (ToF) light-based sensors have been most commonly used to sense tree shape. Karkee and Adhikari [393] described using a ToF camera to capture the shape of an apple tree. Due to the sensor’s limited field of view, a pan-and-tilt stage captures multiple images combined in a data pre-processing step. Image processing and skeletonization steps are carried out, resulting in the tree structure, specifically the trunk and primary branches. This system is also utilized by Karkee et al. [394] to implement a pruning protocol based on branch spacing and branch length. In contrast, Medeiros et al. [395] used a laser detection and ranging sensor, or lidar, to estimate tree shape for a pruning application.

### 8.4. Canopy Management and Training: Influence on Fruit Quality

Many interconnected factors influence fruit quality; training systems are crucial in impacting yield and fruit quality [396]. These systems are closely linked to the environment, tree densities [374,397], and pruning methods [398]. According to Dallabetta et al. [399], fruit quality depends on the cultivar, training system, and position in the canopy. Therefore, several experiments have been conducted to evaluate the role of preharvest factors, such as training systems, on fruit quality [400]. The impact of tree architecture on fruit color development and quality parameters, including soluble solids concentration, titratable acidity, and dry matter, is widely recognized. Additionally, studies have reported the influence of branch fruiting position on fruit quality [401].

Fruit size and color development are critical factors influencing fruit quality, particularly from the consumer’s perspective [402]. Additionally, specific apple cultivars exhibit susceptibility to postharvest disorders, resulting in significant losses of marketable fruit [403]. Zhang et al. [404] identified notable variability in fruit quality associated with canopy position in extensive pear trees. Variability in canopy position has also been documented in figs [405] and peaches [406]. Much of this variability has been linked to light penetration into the canopy [404], showing that fruit quality depends on light absorption, and light is directly related to fruit tree yield [407,408]. Moreover, the combined impact of the training system and photosynthetic photon flux density is also crucial for determining fruit quality [409,410,411]. Another essential factor affecting fruit quality is light quality; for example, red light (600–700 nm) enhances anthocyanin synthesis in fruit peel [412]. The light interception was greater in guava trees planted at wider spacing, and it decreased significantly with the depth of the canopies as planting densities were assessed [413]. With low light interception at higher planting densities, both fruit yield and the consistency of guava fruits decreased [414]. Adequate light distribution within the tree canopy is essential for achieving optimal fruit quality. Large, dense, vertical tree structures restrict light interception, reducing fruit quality characteristics such as yield, fruit weight, color, soluble solids content, and acidity. Moreover, interior canopy positions have demonstrated greater sensitivity to chilling injury disorders in peaches [415]. In the case of apples, adopting high-density orchard systems has resulted in simpler tree canopies with more consistent light penetration [416]. These high-density orchard systems were designed to enhance labor efficiency, productivity, and uniformity in fruit quality. Greater light interception and smaller canopies contribute to increased carbohydrate production allocated to developing fruit [416]. More uniform canopy light environments enhance fruit quality uniformity [402]. Therefore, achieving high yields and fruit quality relies on effective light distribution and substantial light interception within the canopy. Planting at high tree densities, coupled with thin canopy depths as seen in slender spindle and vertical trellis systems, is recognized for yielding positive results [396]. Efficient management practices, including proper pruning, contribute to high yields and good quality, ensuring the maintenance of tree vigor and growth control. In summary, combining modern canopy management and innovative training systems represents a holistic approach to orchard management, aiming to achieve optimal productivity and superior fruit quality [357]. Nevertheless, most studies have predominantly concentrated on vegetative growth and fruiting behavior, with only superficial attention given to fruit quality aspects such as shape, size, total soluble solids, acidity, and aroma [353,417,418]. However, beyond these aspects, there is a growing consumer interest in the health benefits of fruits, including carotenoids, phenolic compounds, and antioxidant capacity [419,420]. These components are closely associated with the postharvest physiology of mature or ripened fruits [421].

## 9. Summary and Future Recommendations

In light of the upcoming challenges of climate change, fruit growers need to be proactive, adapting their practices accordingly. By embracing technologies like precision agriculture and artificial intelligence to monitor and manage microclimate conditions in their orchards, as well as incorporating strategies like shade netting and selecting cultivars that are tolerant to heat or frost, growers can help minimize the impact of climate change on fruit crop yields. Additionally, employing innovative techniques such as biofertilizers to address soil nutrient deficiencies, implementing integrated pest management practices, and utilizing tools like PGRs and film sprays are essential for maintaining the health and quality of fruit crops.

Advancements in genetic editing technologies offer exciting possibilities for enhancing fruit crop traits to meet consumer preferences and increase commercial success. However, further studies on these technologies’ potential benefits and risks in fruit crop improvement are needed.

Canopy management and innovative training systems should also be prioritized to maximize yield efficiency and fruit quality in orchards. Furthermore, creating habitats to support pollinators is essential for sustaining food systems and ensuring successful fruit crop production.

In conclusion, future fruit crop production will undoubtedly be shaped by advancements in innovative technologies, sustainable practices, and research efforts. By investing in these areas, fruit growers can stay ahead of industry challenges and opportunities, ultimately leading to increased success and profitability.

## Figures and Tables

**Figure 1 plants-13-01942-f001:**
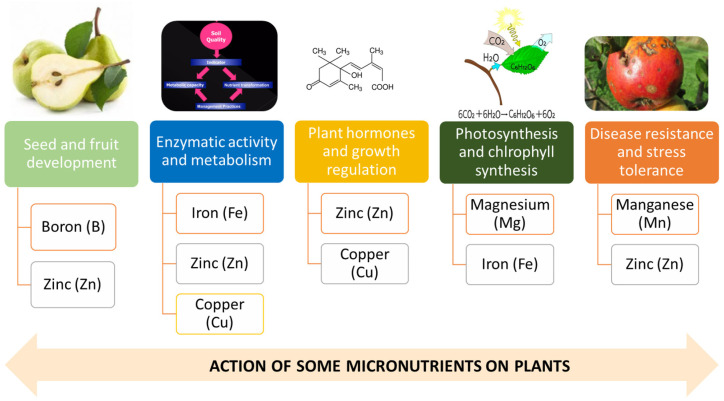
The action of some micronutrients on plants (adapted from Ahmed et al. [28]).

**Figure 2 plants-13-01942-f002:**
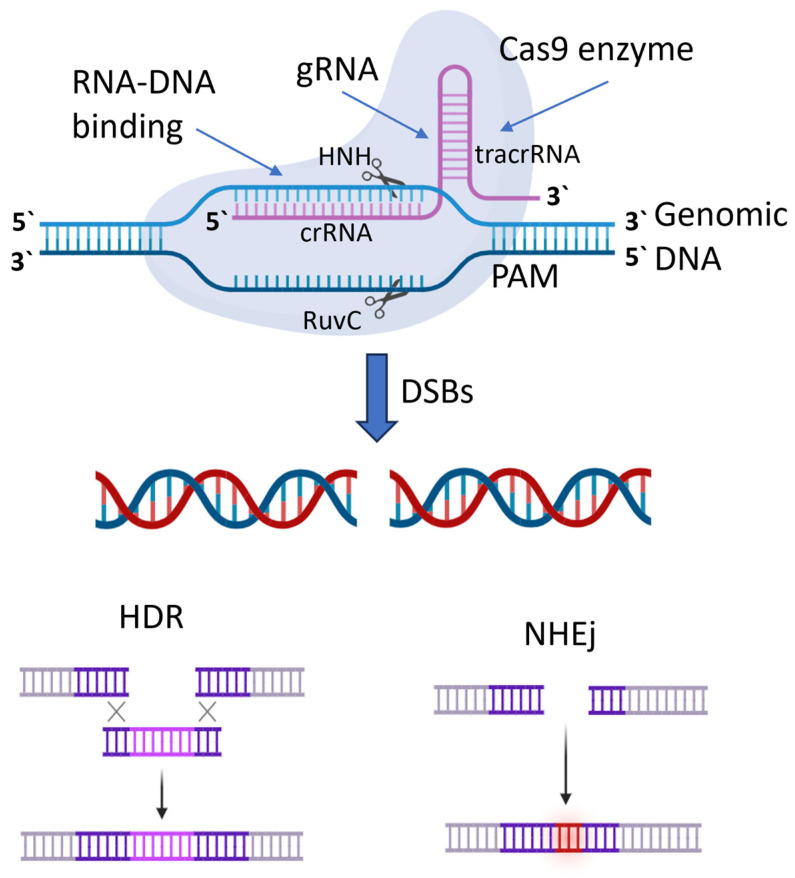
Schematic representation of CRISPR/Cas9 gene editing technique.

**Table 1 plants-13-01942-t001:** Some micronutrient effects on fruit crops.

Micronutrient	Plant Growth	Quality Attributes	Effects	Fruit	Ref.
Boron	Facilitates cell division and sugar transport	Improves fruit firmness, flavor, and nutrient content	Increased fruit set, quality, and seed production	Sweet orange, apple	[50,51,52,53]
Copper	Stimulates enzyme activity and carbohydrate metabolism	It enhances bioactive compounds and improves taste and color	It enhances bioactive compounds and improves taste and color	Citrus	[54]
Iron	Promotes the chlorophyll and the growth of roots	Enhances fruit color, flavor, and nutritional value	Improved root development, reduced chlorophyll synthesis degradation	Cherry, strawberry, citrus	[55,56]
Manganese	Activates enzymes involved in photosynthesis and nutrient metabolism	Improves fruit flavor, sugar, flavonoid content, and storage quality	Increased disease resistance crop yield and biomass accumulation	Citrus, grapes	[57,58,59]
Zinc	Encourages shoot lengthening and hormone regulation	Enhances fruit size, color, and nutritional content	Improved postharvest shelf life and increased pest resistance	Peach	[60]

**Table 2 plants-13-01942-t002:** Advancements in genetic engineering and molecular techniques.

Molecular Techniques	Main Characteristics	Ref.
CRISPR-Cas9	Targeted modifications in the DNA; Efficiency and accuracy of genetic modifications; Incorporation of particular characteristics without the introduction of external genes.	[293,294]
CRISPR (Cas12, Cas13, and Cas 14)	Targeting RNA for gene regulation and providing options for precise genetic modifications.	[295]
High-throughput sequencing and omics technologies	Large-scale analysis of genomes, transcriptomes, and proteomes; Cost-effective and accessible; Aids in the selection of superior breeding candidates.	[296]
MAS	Molecular markers linked to specific genes or traits for selection in breeding programs; Development of high-density marker platforms; Utilization of advanced statistical methods; Rapid identification of plants carrying desired traits.	[297]
Genomic selection	Predicting plants’ performance based on the entire genome; Genomic prediction models due to computational biology and machine learning; Studying complex traits influenced by multiple genes.	[298]
Synthetic biology and gene synthesis	Design and construction of new biological parts; Creation of synthetic genes or pathways; Engineering of crops with novel traits or the optimization of existing pathways.	[299]
RNA interference (RNAi) and gene silencing	Silence or downregulate specific genes through the introduction of small RNA molecules; Targeting gene silencing for controlling pests and diseases.	[300]
Gene stacking and multigenic approaches	Combining multiple genes or traits into a single crop variety (resistance to pests or diseases and tolerance to environmental stresses).	[301]

## Data Availability

Data are contained within the article.

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
