# Peer review of "Impacts of Climate Change and Mitigation Strategies for Some Abiotic and Biotic Constraints Influencing Fruit Growth and Quality"

_plants, 2024, doi:10.3390/plants13141942_

Round 1

Reviewer 1 Report

Comments and Suggestions for Authors

General overview

The review entitled “Impacts and Mitigation Strategies for Abiotic and Biotic Constraints Influencing Fruit Growth and Quality” by Bacelar et. al aims to cover, without a doubt, a relevant topic for agriculture, such as the influence of abiotic and biotic factors that can affect the growth and quality of fruits. As well as management strategies and technologies to face them in a climate change context. Nevertheless, it does not describe the topic in an orderly and clear way for the reader, nor does it go into the form and magnitude in which these factors would be affected and therefore affect the physiology of the crop (focus on fruit growth and quality), often remaining in phrases such as essential, crucial, determining, etc., without making reference to other studies or fruit crop species, which is what the reader expects in a review and update of the topic. It tries to address too many topics and spreads the information too thinly, leaving the focus of the review. Throughout the manuscript, many sentences of minor relevance are repeated, even in some cases in the same paragraphs.

I suggest amending the title or focusing the review on a less generalist aspect.

I encourage to the authors to follow a clear guideline of the topics and consider the work of 2018: Biotic and Abiotic Stress Tolerance in Plants https://doi.org/10.1007/978-981-10-9029-5

My decision was to reject the review, as I consider that there are significant aspects that need to be amended, which would enrich the text and lead to a better review article.

The following are the parts that I would like to see changed or considered:

Abstract

It needs to be improved and written in a logic order, as well as including the topics that really are covered by the review.

Define the types of stresses to which the crop may be subjected and clearly separate abiotic from biotic stresses and the most suitable strategies to cope with them found in this review.

Although it is a review, it should clearly state what the objective of the review is and why it needs to be updated to indicate new scientific background to the topic.

Lines [22-25]

What do the authors mean by "other natural disasters" and what relevance does it have for the review? Ozone increases and the effect over fruits are not discussed in the text.

Lines [25-27]

Not discussed in the review.

Line [36]

Please avoid repeating words from the title of the manuscript.

Sections 1 to 8

Please maintain only information related to the work. Subsection 5.1 is a good example of how to address this issue.

There are some paragraphs without references.

Please avoid starting paragraphs with abbreviations.

The term sustainable in an environmental perspective is more correct that environmentally friendly.

The abbreviation of a term must be made in the first appearance of the manuscript.

Especially in sections 2 to 4, the strategies defined as innovative are really a compilation of some work, which has been studied at least in the last decade. Section 7 is a good example of how to include new techniques in the review.

Please avoid repeating definitions in different paragraphs to provide a more comprehensible reading order for the reader.

Please add the effect of deficit irrigation strategies on fruit growth and quality. Please explore technologies to cope with abiotic stress, such as monitoring and prediction of climatic events that may affect crops, technologies to cope with water scarcity, or biotics, such as remote pest identification.

Please always use formal writing.

There are many words with formatting (font) errors.

1. Introduction

The introduction should clearly define the concept of biotic and abiotic stress and the main parameters of fruit quality (physical, functional, nutritional, and post-harvest suitability) that are affected due to stress.

Lines [50-51]

Please add a more precise objective for the review and why it needs to be updated to indicate new scientific background to the topic. Furthermore, it should be in line with the title (fruit growth and quality). This will contribute to the reader's perception of the work.

Lines [52-53]

The effect of climate change is highly referenced in the text, maybe it should be considered to add it in the title.

2. Effects of Extreme Temperatures on Fruit Growth and Quality and Strategies to Mitigate these Impacts

Please consider the title of the Section as "2. Abiotic factors determining fruit growth and quality". The subsections (i.e. 2.1 Temperature) should be each factor and at the end the strategies to cope them, which usually overlap in their effect.

Lines [63]

Please consider: Since 1950, …

Lines [74]

What physiological disorders? in fruit set, fruit growth, harvest, post-harvest, bud differentiation and flower induction? Please give examples.

Lines [79-81]

It should be noted that the length of the growing season and temperatures are likely to influence the crop's water requirement, which will affect the crop's water status especially in arid and semi-arid areas, if they are stressed when the fruit is most sensitive to changes in the rainfall regime or to a reduced availability of water for irrigation.

Lines [81-83]

Introduction.

Line [84]

The same idea in line 77, please re-order this section.

Line [87]

Please consider “radiation”.

Line [88-89]

It is important to mention at least in which group of fruit crops it occurs, as some are not as susceptible.

Line [92]

Please define as plant growth regulators (PGRs).

Line [92-93]

Please consider: …rootstocks and fruit cultivars, application of growth regulators, and the use of bagging and film sprays [9]. On the other side, Despite the inevitable and permanent consequences of global warming, fruit crops are particularly vulnerable to extreme cold temperatures and frost damage.

Lines [91-93]

What new relevant information is referenced in addition to the Sharma et al. review?

Lines [94-102]

Please elaborate further, the entire paragraph refers to a single report.

What new relevant information is referenced in addition to the Lavelle et al.?

Line [97]

…and disrupt fruit development by delaying flowering and fruit sets. It can…

Line [99]

…such as using frost blankets, modifying microclimate,, selection…

Line [100]

Please use PGRs and give some examples of PGRs and fruit crops.

Lines [112-113]

The entire process of pollination and fecundation is critical for fruit set. Temperature can affect stigma receptivity, decrease flower quality, inhibit pollen tube development, etc. What happens with anemophilous pollinated fruit trees?

Line [114]

Please add: …premium fruits. In addition, extreme temperatures can negatively affect the activity…

Line [121-124]

It is important to mention that in addition to crop susceptibility, all crops can be subject to frost damage, given the intensity, duration and phenological stage at which the crop is subjected to frost, including grapevine.

3. Influence of Soil Nutrient Deficiencies or Imbalances on Fruit Growth and Quality and Innovative Approaches to Improve Soil Fertility

Please rewrite this section. The paragraphs provide limited information on the effect of soil quality on growth and fruit quality. Which fruits physiological disorders are influenced by soil properties, e.g. Ca, Ca/K, Ca/Mg?

Define the abbreviation in the first occurrence of the text and then use it. Paragraphs are repeated. There is information irrelevant to the topic and no in-depth information about the fruit.

The table 1 provides more relevant information than the text.

Line [128-137]

This paragraph should be removed or moved to the introduction, as it deviates from the objectives of the review.

Line [149-169]

This paragraph should elaborate further on the points made in figure 1 and add important effects that are not mentioned, which is in line with the review, and use the figure as a summary for the reader.

Even in small concentrations quantities, soil nutrients are essential for plant development, maturation, fruit quality, and reproduction. These can be divided into macronutrients and micronutrients, and both play an essential role in metabolic and physiological processes. Macronutrients are necessary for the growth and development of plants in high quantities, and micronutrients are essential in small amounts. Furthermore, micronutrients can cause toxicity problems, impacting the production and quality of crops if they are absorbed (incorrect term) in quantities greater than those necessary for each plant species. Currently, it is considered that there are twenty mineral elements essential or beneficial for the growth and development of plants [18]. According to Aftab and Hakeem [19], because some nutrients are considered “micronutrients,” small amounts of these elements are needed in the soil for the full development of plants and their vitality. When involved in physiological and biochemical processes, these elements contribute to the harmonious development of fruits. Some examples of macronutrients are nitrogen (N), phosphorus (P), potassium (K), calcium (Ca), magnesium (Mg), and sulfur (S) [18,20]. Some micronutrients include iron (Fe), zinc (Zn), manganese (Mn), copper (Cu), molybdenum (Mo), boron (B), nickel (Ni), cobalt (Co), silicon (Si), sodium (Na) and chlorine (Cl). In Figure 1, some micronutrients and their action on plants can be seen.

Lines [170-192]

Please maintain only information related to the work.

Lines [258-260]

Any references?

4. Strategies for Integrated Pest Management, including Biological Control, Cultural Practices, and Minimum Use of Pesticide Applications

The IPM does not cover the entire title?

Please avoid starting paragraphs with abbreviations.

Lines [307-320]

Is it relevant to the review?

The complete paragraph does not correspond to the reference, references are missing.

Lines [329-330]

Please use formal writing.

Lines [345-346]

The same description in the paragraphs above, text repeated.

Lines [347-348]

Please use formal writing.

5. Effects of Light on Fruit Development, Color, and Flavor, and Strategies to Optimize Light Conditions through Pruning, Thinning, and Proper Orchard Layout

Lines [495-526]

There are many words with formatting (font) errors.

Line [557]

Subindex in CO2.

6. Importance of Pollination on Fruit Set and Quality and Strategies to Enhance Pollination

Line [667]

Please use exotic or non-native species.

Line [690]

Please change: … honeybee (Apis mellifera).

Line [697]

The indentation is missing.

Line [712]

Double space after vegetables, …

Line [754]

Please use the scientific name at the first occurrence of the species in the text.

Line [754, 758, 767, 774]

Honeybees or honey-bees?

Line [773]

Please add point after spp.

7. Advancements in Genetic Engineering and Molecular Techniques for Selecting and Breeding Varieties with Desired Traits

Line [783]

Please replace crops with fruit.

Line [786]

Delete the extra “,”.

8. Canopy Management and Training: Influence on Fruit Quality

Please summarize to what is strictly necessary for the purpose of the revision. It is even possible to consider adding in Section 5 which discusses light interception.

Lines [1148-1151]

Reference is missing.

9. Summary and Future Recommendations

Lines [1154-1155]

Please use the term climate change.

Line [1155]

There is no section that discusses the effect of irrigation scheduling on fruit growth and quality. There is a large amount of literature that focuses on deficit irrigation to increase fruit quality, such as sugar accumulation, maturity index, colouring and synthesis of functional compounds. I encourage you to make an in-depth review of this topic in recent years.

Lines [1157]

Please use the term biofertilizer throughout the manuscript, as defined by Sahoo et al.

Lines [1154-1155]

I suggest that the last paragraph should focus on a brief description of the future challenges identified by the authors in the review, to serve as a basis for future research.

References

Please comply with the citation guidelines of the journal.

All journal names should be abbreviated.

Many references do not have a DOI.

Please add a “.” after the DOI.

Comments on the Quality of English Language

English is well written.

Author Response

Response letter to Reviewer #1

Response to Reviewer’s Comments

We thank the reviewer for the detailed evaluation and insightful feedback. We have taken into consideration all the suggestions for improving the clarity and organization of the manuscript and have made the necessary changes to the best of our ability in the new document. We present the detailed responses to them point by point. changes are highlighted in the manuscript.

Detailed responses to a reviewer’s suggestions point by point

I suggest amending the title or focusing the review on a less generalist aspect.

Response: We have changed the title as suggested and added “The effect of climate change” as was also suggested.

The following are the parts that I would like to see changed or considered:

Abstract

It needs to be improved and written in a logic order, as well as including the topics that really are covered by the review.

Response: We agree. The abstract was rewritten in a more logical order, covering all key topics, indicating the objectives of the review.

Lines [22-25]

What do the authors mean by "other natural disasters" and what relevance does it have for the review? Ozone increases and the effect over fruits are not discussed in the text.

Response: The sentence was rewritten, eliminating what was not dicussed in the text.

Lines [25-27]

Not discussed in the review.

Response: We agree and it was eliminated also.

Line [36]

Please avoid repeating words from the title of the manuscript.

Response: We agree. New keywords are presented, more related to the review and not in the title.

Sections 1 to 8

Please maintain only information related to the work. Subsection 5.1 is a good example of how to address this issue.

Response: We agree and we reformulated the text.

There are some paragraphs without references.

Response: We acknowledge that the recommendation and references were introduced when they were previously missing.

Please avoid starting paragraphs with abbreviations.

Response: We agree and have made some modifications according to the instructions of the reviewer.

The term sustainable in an environmental perspective is more correct that environmentally friendly.

Re: The term was replaced, as suggested along the manuscript.

The abbreviation of a term must be made in the first appearance of the manuscript.

Re: We agree and it was done.

Especially in sections 2 to 4, the strategies defined as innovative are really a compilation of some work, which has been studied at least in the last decade. Section 7 is a good example of how to include new techniques in the review.

Please avoid repeating definitions in different paragraphs to provide a more comprehensible reading order for the reader.

Re: Section 2 and 4 were modified.

Please add the effect of deficit irrigation strategies on fruit growth and quality. Please explore technologies to cope with abiotic stress, such as monitoring and prediction of climatic events that may affect crops, technologies to cope with water scarcity, or biotics, such as remote pest identification.

Re: It was partially done. Remote pest identification was included.

Please always use formal writing.

Re: It was done.

There are many words with formatting (font) errors.

Re: It was emended.

  1. Introduction

The introduction should clearly define the concept of biotic and abiotic stress and the main parameters of fruit quality (physical, functional, nutritional, and post-harvest suitability) that are affected due to stress.

Re: It was included in introduction.

Lines [50-51]

Please add a more precise objective for the review and why it needs to be updated to indicate new scientific background to the topic. Furthermore, it should be in line with the title (fruit growth and quality). This will contribute to the reader's perception of the work.

Re: A more precise objective included in introduction.

Lines [52-53]

The effect of climate change is highly referenced in the text, maybe it should be considered to add it in the title.

Re: It was done.

  1. Effects of Extreme Temperatures on Fruit Growth and Quality and Strategies to Mitigate these Impacts

Please consider the title of the Section as "2. Abiotic factors determining fruit growth and quality". The subsections (i.e. 2.1 Temperature) should be each factor and at the end the strategies to cope them, which usually overlap in their effect.

Re: After careful consideration, we have determined that it would not be advantageous to make changes to the structure of the manuscript as proposed, as it may cause confusion at this time.

Lines [63]

Please consider: Since 1950, …

Re: We have done it.

Lines [74]

What physiological disorders? in fruit set, fruit growth, harvest, post-harvest, bud differentiation and flower induction? Please give examples.

Re: Examples were given in the text (“High day and evening temperatures occurring more frequently now due to climate change can cause the drop of flowers and produce deformed or undersized fruits”).

Lines [79-81]

It should be noted that the length of the growing season and temperatures are likely to influence the crop's water requirement, which will affect the crop's water status especially in arid and semi-arid areas, if they are stressed when the fruit is most sensitive to changes in the rainfall regime or to a reduced availability of water for irrigation.

Re: It was introduced.

Lines [81-83]

Introduction.

Line [84]

The same idea in line 77, please re-order this section.

Re: It was done.

Line [87]

Please consider “radiation”.

Re: It was done.

Line [88-89]

It is important to mention at least in which group of fruit crops it occurs, as some are not as susceptible.

Re: Examples were given.

Line [92]

Please define as plant growth regulators (PGRs).

Re: We define PGRs and included the objectives of it use.

Line [92-93]

Please consider: …rootstocks and fruit cultivars, application of growth regulators, and the use of bagging and film sprays [9]. On the other side, Despite the inevitable and permanent consequences of global warming, fruit crops are particularly vulnerable to extreme cold temperatures and frost damage.

Re: We considered the suggestion that we acknowledge.

Lines [91-93]

What new relevant information is referenced in addition to the Sharma et al. review?

Re: More information was introduced.

Lines [94-102]

Please elaborate further, the entire paragraph refers to a single report.

What new relevant information is referenced in addition to the Lavelle et al.?

Line [97]

…and disrupt fruit development by delaying flowering and fruit sets. It can…

Re: Language was improved.

Line [99]

…such as using frost blankets, modifying microclimate,, selection…

Re: It was emended.

Line [100]

Please use PGRs and give some examples of PGRs and fruit crops.

Lines [112-113]

Re: It was done.

The entire process of pollination and fecundation is critical for fruit set. Temperature can affect stigma receptivity, decrease flower quality, inhibit pollen tube development, etc. What happens with anemophilous pollinated fruit trees?

Re: This point is now discussed in the manuscript.

Line [114]

Please add: …premium fruits. In addition, extreme temperatures can negatively affect the activity…

Re: We add it.

Line [121-124]

It is important to mention that in addition to crop susceptibility, all crops can be subject to frost damage, given the intensity, duration and phenological stage at which the crop is subjected to frost, including grapevine.

Re: It was taken in consideration.

  1. Influence of Soil Nutrient Deficiencies or Imbalances on Fruit Growth and Quality and Innovative Approaches to Improve Soil Fertility

Please rewrite this section. The paragraphs provide limited information on the effect of soil quality on growth and fruit quality. Which fruits physiological disorders are influenced by soil properties, e.g. Ca, Ca/K, Ca/Mg?

Re: We appreciate the comments of the reviewer as it allowed us to improve the document by responding to his suggestions. Thus, we understand the reviewer's concern and the document has been improved according to reviewers' recommendations.

Regarding the question posed in this comment about the physiological disorders of these elements, the answer appears throughout the new version.

Define the abbreviation in the first occurrence of the text and then use it. Paragraphs are repeated. There is information irrelevant to the topic and no in-depth information about the fruit.

Re: We agree and have made some modifications according to the instructions of the reviewer. However, no more information about the fruit was added in this section as this is intended to mainly discuss the influence of soil nutrient deficiencies or imbalances on fruit growth and quality.

The table 1 provides more relevant information than the text.

Re: The authors understand the reviewer's concern. However, it is important to note that no more information was included in the text already included in the table to avoid redundancy.

Line [128-137]

This paragraph should be removed or moved to the introduction, as it deviates from the objectives of the review.

Re: The paragraph was removed.

Line [149-169]

This paragraph should elaborate further on the points made in figure 1 and add important effects that are not mentioned, which is in line with the review, and use the figure as a summary for the reader.

Even in small concentrations quantities, soil nutrients are essential for plant development, maturation, fruit quality, and reproduction. These can be divided into macronutrients and micronutrients, and both play an essential role in metabolic and physiological processes. Macronutrients are necessary for the growth and development of plants in high quantities, and micronutrients are essential in small amounts. Furthermore, micronutrients can cause toxicity problems, impacting the production and quality of crops if they are absorbed (incorrect term) in quantities greater than those necessary for each plant species. Currently, it is considered that there are twenty mineral elements essential or beneficial for the growth and development of plants [18]. According to Aftab and Hakeem [19], because some nutrients are considered “micronutrients,” small amounts of these elements are needed in the soil for the full development of plants and their vitality. When involved in physiological and biochemical processes, these elements contribute to the harmonious development of fruits. Some examples of macronutrients are nitrogen (N), phosphorus (P), potassium (K), calcium (Ca), magnesium (Mg), and sulfur (S) [18,20]. Some micronutrients include iron (Fe), zinc (Zn), manganese (Mn), copper (Cu), molybdenum (Mo), boron (B), nickel (Ni), cobalt (Co), silicon (Si), sodium (Na) and chlorine (Cl). In Figure 1, some micronutrients and their action on plants can be seen.

Re: We agreed with the reviewer. Effectively, figure 1 should summarize what is discussed in depth in the text. Thus, as figure 1 summarizes the action of some micronutrients on plants, it was moved to the "Micronutrient" section, where the effects of each micronutrient on plants are mentioned in greater detail.

Lines [170-192]

Please maintain only information related to the work.

Re: We agreed with the reviewer and changes were made.

Lines [258-260]

Any references?

Re: We agreed with the reviewer, it was a mistake, and a reference was added.

  1. Strategies for Integrated Pest Management, including Biological Control, Cultural Practices, and Minimum Use of Pesticide Applications

The IPM does not cover the entire title?

Re: We agree. The title was modified to Strategies for Integrated Pest Management.

Please avoid starting paragraphs with abbreviations.

Re: We agree. The abbreviations were all revised.

Lines [307-320]

Is it relevant to the review?

The complete paragraph does not correspond to the reference, references are missing.

Re: Yes, it is relevant to the review, the references were added at the end of all sentences.

Lines [329-330]

Please use formal writing.

Re: We agree, and for this reason, the sentence was rewritten.

Lines [345-346]

The same description in the paragraphs above, text repeated.

Re: We agree, and for this reason, we removed the sentence.

Lines [347-348]

Please use formal writing.

Re: We agree, and for this reason, the sentence was rewritten.

  1. Effects of Light on Fruit Development, Color, and Flavor, and Strategies to Optimize Light Conditions through Pruning, Thinning, and Proper Orchard Layout

Lines [495-526]

There are many words with formatting (font) errors.

Re: We agree. The whole sentence was revised.

Line [557]

Subindex in CO2.

Re: Thank you. The error was corrected.

  1. Importance of Pollination on Fruit Set and Quality and Strategies to Enhance Pollination

Line [667]

Please use exotic or non-native species.

Re: “Non native species” was used in the sentence.

Line [690]

Please change: … honeybee (Apis mellifera).

Re: The sentence was changed accordingly as suggested

Line [697]

The indentation is missing.

Re: Indentation was added.

Line [712]

Double space after vegetables, …

Re: The double spacing was corrected to single spacing.

Line [754]

Please use the scientific name at the first occurrence of the species in the text.

Re: Scientific name was added.

Line [754, 758, 767, 774]

Honeybees or honey-bees?

Re: The terminology was uniformed in the text.

Line [773]

Please add point after spp.

Re: The missing point was added.

  1. Advancements in Genetic Engineering and Molecular Techniques for Selecting and Breeding Varieties with Desired Traits

Line [783]

Please replace crops with fruit.

Re: The word “crop” was replaced by the word “fruits”.

Line [786]

Delete the extra “,”.

Re: It was deleted.

  1. Canopy Management and Training: Influence on Fruit Quality

Please summarize to what is strictly necessary for the purpose of the revision. It is even possible to consider adding in Section 5 which discusses light interception.

Re: As suggested by the reviewer it was summarized what is strictly necessary for the purpose of the revision and the changes are highlighted in the manuscript.

Lines [1148-1151]

Reference is missing.

Re: As suggested by the reviewer the missing references were introduced in the manuscript and in the reference list of the manuscript.

  1. Summary and Future Recommendations

Lines [1154-1155]

Please use the term climate change.

Re: Done.

Line [1155]

There is no section that discusses the effect of irrigation scheduling on fruit growth and quality. There is a large amount of literature that focuses on deficit irrigation to increase fruit quality, such as sugar accumulation, maturity index, colouring and synthesis of functional compounds. I encourage you to make an in-depth review of this topic in recent years.

Re: It was removed since it was not discussed.

Lines [1157]

Please use the term biofertilizer throughout the manuscript, as defined by Sahoo et al.

Re: Done.

Lines [1154-1155]

I suggest that the last paragraph should focus on a brief description of the future challenges identified by the authors in the review, to serve as a basis for future research.

Re: It was rewritten.

References

Please comply with the citation guidelines of the journal.

All journal names should be abbreviated.

Many references do not have a DOI.

Please add a “.” after the DOI.

Reference list was revised.

Reviewer 2 Report

Comments and Suggestions for Authors

This manuscript is a pervasive and informative review of methods to alleviate constraints influencing fruit growth and quality. Without prejudice to the above, some problems must be addressed, mainly concerning the correct reference to crucial statements. On the other hand, at some moments, the study objective is not clear. Is it plants or fruits? Of course, it is not easy to separate, but the review is about fruit, so it is necessary to point the focus.

In the following lines, I will resume my comments and questions:

The abstract did not give an appropriate final statement about, for example, future recommendations.

The abstract is centered on fruit development problems, but this work could easily be attributed to the whole plant, so it is imperative to differentiate the focus.

I miss some things like melatonin treatments, foliar spraying techniques, and IA for modeling and proposing solutions, as appropriate according to each chapter.

L59. I’m not sure that citing Chmielewski is suitable for the first paragraph, as that paper is only about German reality.

L73. "Damage to macromolecules, altered gene expression, affected membrane fluidity": what is the reference? Indeed, these terms are no longer available unless they are essential, as the same text stated.

L94 to L98. Reference?

L107 to L110. Reference?

L150/152. How is it defined as a micro or a micronutrient? (micro- or milligrams vs grams requirement)

Fig1. Correct “developmente”

L178. There are font issues like “of the fruit” in this line, but not only here, especially in chapter 5.

L178. How could it be produced an “excess of nitrogen”? Please refer to agronomic practices and natural events that can change the N concentration, as mentioned for other nutrients.

L183. What are the major molecular functions of P?

L201. Why is calcium so important? I miss an explanation about calcium as a second messenger.

L212. What does “low fruit conservation capacity” mean?

Table 1. Sort alphabetically or in a comprehensive way.

L307-318. References?

L340. Take care of using the “you” pronoun that is present many times in chapter 4.

L365-366. Please give more references to this sentence or modify it because, as far as I know, it is true inside the European Union (EU) because of the  EU Directive Sustainable Use (Directive 2009/128/EC).

L379. CBC and ABC acronyms were defined previously.

L429. “demonstrated in this experiment.” To what experiment did you refer?

L790-819. Maybe you can present a diagram or a table of each one with keynotes.

L430-434. References?

L455-458. Maybe a list of commonly used synthetic and natural pesticides with some advantages and disadvantages.

L474. “Pesticides play a significant role in maintaining many terrible diseases”? 

L478. “The presence of pesticide residues in different crops has hurt the export of agricultural products.” Reference?

L967. What’s the difference between canopy management and orchard design, pruning, and thinning, as mentioned in Chapter 5?

Some ideas are sometimes repetitive.

Comments on the Quality of English Language

I have mentioned a few issues in the comments.

Author Response

Response letter to Reviewer #2

Response to Reviewer’s Comments

Comments and Suggestions for Authors

This manuscript is a pervasive and informative review of methods to alleviate constraints influencing fruit growth and quality. Without prejudice to the above, some problems must be addressed, mainly concerning the correct reference to crucial statements. On the other hand, at some moments, the study objective is not clear. Is it plants or fruits? Of course, it is not easy to separate, but the review is about fruit, so it is necessary to point the focus.

Response: We agree and we thank the reviewer for the detailed evaluation of the manuscript. We focus on fruit growth and quality and clearly differentiate between plants and fruits in the revised version of the review. We appreciate your insights on the importance of referencing crucial statements and clarifying the study objective. Thank you for pointing out these areas for improvement. and insightful feedback. We have taken into consideration all the suggestions for improving the clarity and organization of the manuscript and have made the necessary changes to the best of our ability in the new document. We present the detailed responses to them point by point. Changes are highlighted in the manuscript.

In the following lines, I will resume my comments and questions:

The abstract did not give an appropriate final statement about, for example, future recommendations.

The abstract is centered on fruit development problems, but this work could easily be attributed to the whole plant, so it is imperative to differentiate the focus.

Response: We agree. The abstract was rewritten and recommendations have been taken in account.

I miss some things like melatonin treatments, foliar spraying techniques, and IA for modeling and proposing solutions, as appropriate according to each chapter.

Response: We have introduced new information about the subject.

L59. I’m not sure that citing Chmielewski is suitable for the first paragraph, as that paper is only about German reality.

Response: We have instead introduced the book by Taiz, L.; Zeiger, E.; Møller, I.M.; Murphy, A.S. Fundamentals of Plant Physiology, which offers a broader perspective on the topic. Thank you for bringing this to our attention.

L73. "Damage to macromolecules, altered gene expression, affected membrane fluidity": what is the reference? Indeed, these terms are no longer available unless they are essential, as the same text stated.

Response: We agree and eliminated that in the text.

L94 to L98. Reference?          

Response:  This section was partially rewritten.

L107 to L110. Reference?

Response: This section was partially rewritten.

L150/152. How is it defined as a micro or a micronutrient? (micro- or milligrams vs grams requirement)

Response: According Tariq Aftab, 2020, micronutrients are essential elements that plants require in minimal quantities for optimal growth, development, and reproduction. Although these quantities are relatively small, they are indispensable for various physiological and metabolic processes. Some primary elements classified as micronutrients include iron (Fe), zinc (Zn), manganese (Mn), copper (Cu), molybdenum (Mo), boron (B), and chloride (Cl). These micronutrients are so named because they are needed in much smaller or ‘trace’ amounts and weighed in milligrams (1/1,000 gram) and micrograms (1/1,000,000 gram).

K.R.H. Tariq Aftabv. Plant Micronutrients: Deficiency and Toxicity Management, Springer (2020). https://doi.org/10.1007/978-3-030-49856-6

Fig1. Correct “developmente”

Response: Done.

L178. There are font issues like “of the fruit” in this line, but not only here, especially in chapter 5.

Response: Done.

L178. How could it be produced an “excess of nitrogen”? Please refer to agronomic practices and natural events that can change the N concentration, as mentioned for other nutrients.

Response: To answer this question, it was added in the manuscript: “The nature and degree of interactive impacts of management practices and site conditions on N use efficiency are not yet well understood, limiting a comprehensive assessment of these practices on N use efficiency. Implementing optimal agricultural management strategies to increase global N utilization efficiency (48%) is urgent to ensure environmental safety and benefits (Quan et al., 2021). According to You et al. (2023), in all cases studied, nutrient and crop management practices increased N use efficiency, while soil management showed the opposite impact and decreased N use efficiency, including crop type, soil pH, soil clay content, soil organic carbon, temperature and precipitation affected N use efficiency.”

L183. What are the major molecular functions of P?

Response: To answer this question, it was added in the manuscript: “Phosphorus intervenes in almost all plant growth and metabolism processes. Thus, it is observed that phosphorus is translocated to areas with high necessity of energy, such as the fruiting areas of the plant, for the formation of seeds and fruits.”

L201. Why is calcium so important? I miss an explanation about calcium as a second messenger.

Response: We have discussed the importance of Ca in the new version (Lines 209-228).

L212. What does “low fruit conservation capacity” mean?

Response: The sentence was modified to: “leading to reduced fruit growth and shelf life”.

Table 1. Sort alphabetically or in a comprehensive way.

Response: The table has been amended. Now the micronutrients appear in alphabetical order.

L307-318. References?

Response: Yes, it is relevant to the review, the references were added at the end of all sentences.

L340. Take care of using the “you” pronoun that is present many times in chapter 4.

Response: We agree, therefore, that the sentences using the pronoun "you" have been rewritten.

L365-366. Please give more references to this sentence or modify it because, as far as I know, it is true inside the European Union (EU) because of the  EU Directive Sustainable Use (Directive 2009/128/EC).

Response: The references were added at the end of the sentence.

L379. CBC and ABC acronyms were defined previously.

Response: We agree, we revised all the acronyms.

L429. “demonstrated in this experiment.” To what experiment did you refer?

Response: The sentence has been changed according to the reference.

L430-434. References?

Response: Yes, it is relevant to the review, the references were added at the end of all sentences.

L455-458. Maybe a list of commonly used synthetic and natural pesticides with some advantages and disadvantages.

Response: Due to the length of the article and its status as just a suggestion, the list was not made.

L474. “Pesticides play a significant role in maintaining many terrible diseases”?

Response: We agree, and for this reason, we removed the sentence.

L478. “The presence of pesticide residues in different crops has hurt the export of agricultural products.” Reference?

Response: The reference was added at the end of the sentence.

L790-819. Maybe you can present a diagram or a table of each one with keynotes.

Response: We agree with the suggestion. A new table (table 2) was introduced.

L967. What’s the difference between canopy management and orchard design, pruning, and thinning, as mentioned in Chapter 5?

Response: While canopy management is a specific aspect of orchard management that focuses on the manipulation of the canopy itself, orchard design, pruning, and thinning encompass a broader range of practices that contribute to the overall health and productivity of the orchard. We have clarified that in the new version of the manuscript.

Some ideas are sometimes repetitive.

Response: We appreciate the input and tried to avoid repetitive ideas.

Round 2

Reviewer 1 Report

Comments and Suggestions for Authors

I thank the authors for having considered many of my recommendations. I believe that the quality of the manuscript has been significantly improved. Nevertheless, there are still too many typos and formatting errors that need to be amended before publication and please, before sending them to the review process! (at this point the review should be focused on substantive issues and not on form).  Also, in sections 1, 2 and 3 there are several sentences that do not have their references, please add them. Finally, I strongly request the authors to read the text completely before resubmitting it, to avoid redundancy of paragraphs that state the same point and if necessary reorder them. All my comments are in the attached pdf. Best regards.

Comments on the Quality of English Language

Please avoid repeating a sentence or definition too many times in the same paragraph.

Author Response

Dear Reviewer 1,

We would like to express our gratitude to review our document once again and for providing valuable suggestions and comments. We have made the necessary revisions to the document based on your feedback. The new version of the document has been sent to you with the new changes highlighted in green, as well as the new references (to be included) at the beginning of the reference list. Additionally, we have also uploaded the responses to your comments in the attached PDF file for your reference.

Thank you once again for your great effort in helping us improve our work.

Best regards

Round 3

Reviewer 1 Report

Comments and Suggestions for Authors

I thank the authors for addressing most of my suggestions. Nevertheless, there are still some minor considerations which are listed below and, once amended, I believe the manuscript is acceptable for publication. Thank you to all your team for their efforts and receive my best regards.

Line 95:  ...(discussed in insection 5)

Line 107: ...artificial intelligence 
(IA) -> AI

Line 149: including grapevine[new ref-

Line 253: [36,37]. Furthermore, micronutrients, such as Z and Mg are essential in small quantities for various...

Line 337-342: Please change to

The Integrated Pest Management (IPM) is an effective process that can control pests while minimizing environmental and human risks and is adaptable to environments, including urban, agricultural, wild, or natural areas [70]. Nevertheless, this crop protection solution is far from universal– it is often tricky, indeed sometimes impossible to implement, due to several factors as pest dynamics, host-plant and climate interactions, practicalities of crop production, and socioeconomic conditions in the region of interest, that can hinder the full implementation of this strategy [69]. 

Line 347:  approach to pest control, which is both sustainable and safe. [70]. 

Line 461 (from review 2): requires collective action, according to Baker] [86]. 

Line 614 (from review 2): Please add the blueberry cultivar or delete "cultivar"
.... apple [160], 
blueberry cultivar [158], peach [161], orange....

Lines 618-620: Please change to

Al-Saif et al [162] found that different pruning intensities performed on 'Valencia' orange trees positively affected fruit quality traits such as size, firmness, juice content, total soluble solids (TSS), TSS/acid ratio and vitamin C content. 

Author Response

We carefully addressed the comments/suggestions of Reviewer 1 and made the necessary revisions to the manuscript. We are resubmitting the revised version with changes highlighted in blue. We hope that the manuscript is now suitable for publication in Plants.

Thank you in advance. Our best regards,